# oViT: An Accurate Second-Order Pruning Framework for Vision Transformers

## Abstract

Models from the Vision Transformer (ViT) family have recently provided breakthrough results across image classification tasks such as ImageNet. Yet, they still face barriers to deployment, notably the fact that their accuracy can be severely impacted by compression techniques such as pruning. In this paper, we take a step towards addressing this issue by introducing *Optimal ViT Surgeon (oViT)*, a new state-of-the-art weight sparsification method, which is particularly well-suited to Vision Transformers (ViT) models. At the technical level, oViT introduces a new weight pruning algorithm which leverages second-order information, and in particular can handle weight correlations accurately and efficiently. We complement this accurate one-shot pruner with an in-depth investigation of gradual pruning, augmentation, and recovery schedules for ViTs, which we show to be critical for successful compression. We validate our method via extensive experiments on classical ViT and DeiT models, hybrid architectures, such as XCiT, EfficientFormer and Swin, as well as general models, such as highly-accurate ResNet and EfficientNet variants. Our results show for the first time that ViT-family models can in fact be pruned to high sparsity levels (e.g. $\geq 75\%$) with low impact on accuracy ($\leq 1\%$ relative drop). In addition, we show that our method is compatible with structured pruning methods and quantization, and that it can lead to significant speedups on a sparsity-aware inference engine.

## 1 Introduction

Attention-based Transformers (Vaswani et al., 2017) have revolutionized natural language processing (NLP), and have become popular recently also in computer vision (Dosovitskiy et al., 2020; Touvron et al., 2021; Carion et al., 2020). The Vision Transformer (ViT) (Dosovitskiy et al., 2020; Touvron et al., 2021) and its extensions (Ali et al., 2021; Liu et al., 2021; Wang et al., 2021) which are the focus of our study, have been remarkably successful, despite encoding fewer inductive biases. However, the high accuracy of ViTs comes at the cost of large computational and parameter budgets. In particular, ViT models are well-known to be more parameter-heavy (Dosovitskiy et al., 2020; Touvron et al., 2021), relative to their convolutional counterparts. Consequently, a rapidly-expanding line of work has been focusing on reducing these costs for ViT models via *model compression*, thus enabling their deployment in resource-constrained settings.

Several recent references adapted compression approaches to ViT models, investigating either structured pruning, removing patches or tokens, or unstructured pruning, removing weights. The consensus in the literature is that ViT models are generally less compressible than convolutional networks (CNNs) at the same accuracy. If the classic ResNet50 model (He et al., 2016) can be compressed to 80-90% sparsity with negligible loss of accuracy, e.g. (Frantar et al., 2021; Peste et al., 2021), the best currently-known results for similarly-accurate ViT models can only reach at most 50% sparsity while maintaining dense accuracy (Chen et al., 2021). It is therefore natural to ask whether this "lack of compressibility" is an inherent limitation of ViTs, or whether better results can be obtained via improved compression methods designed for these architectures.

**Contributions.** In this paper, we propose a new pruning method called *Optimal ViT Surgeon (oViT)* , which improves the state-of-the-art accuracy-vs-sparsity trade-off for ViT family models, and shows that they can be pruned to similar levels as CNNs. Our work is based on an in-depth investigation of ViT performance under pruning, and provides contributions across three main directions:

- **A new second-order sparse projection.** To address the fact that ViTs tend to lose significant accuracy upon each pruning step, we introduce a novel approximate second-order pruner called oViT, inspired by the classical second-order OBS framework (Hassibi et al., 1993). The key new feature of our pruner is that, for the first time, it can handle weight correlations

during pruning both accurately and efficiently, by a new theoretical result which reduces weight selection to a sparse regression problem. This approach leads to state-of-the-art results one-shot pruning for both ViTs and conventional models (e.g. ResNets).

- **Post-pruning recovery framework.** To address the issue that ViTs are notoriously hard to train and fine-tune (Touvron et al., 2021; Steiner et al., 2021), we provide a set of efficient *sparse fine-tuning recipes*, enabling accuracy recovery at reasonable computational budgets.

- **End-to-end framework for sparsity sweeps.** Our accurate oViT pruner enables us to avoid the standard and computationally-heavy procedure of gradual pruning for every sparsity target independently, e.g. (Gale et al., 2019; Singh & Alistarh, 2020). Instead, we propose a simple pruning framework that produces sparse accurate models for a sequence of sparsity targets *in a single run*, accommodating various deployments under a fixed compute budget.

Our experiments focus on the standard ImageNet-1K benchmark (Russakovsky et al., 2015). We show that, under low fine-tuning budgets, the oViT approach matches or improves upon the state-of-the-art SViTE (Chen et al., 2021) unstructured method at low-to-medium (40-50%) sparsities, and significantly outperforms it at higher sparsities ($\geq 60\%$) required to obtain inference speedups. Specifically, our results show that, at low targets (e.g. 40-50%), sparsity acts as a regularizer, sometimes *improving* the validation accuracy relative to the dense baseline, by margins between 0.5% and 1.8% Top-1 accuracy. At the same time, we show for the first time that ViT models can attain high sparsity levels without significant accuracy impact: specifically, we can achieve 75-80% sparsity with relatively minor (<1%) accuracy loss. Figure 1 summarizes our results.

Conceptually, we show that ViT models *do not require* over-parametrization to achieve high accuracy, and that, post-pruning, they can be competitive with residual networks in terms of accuracy-per-parameter. Practically, we show that the resulting sparse ViTs can be executed with speedups on a sparsity-aware inference engine (Kurtz et al., 2020). In addition, oViT is complementary to orthogonal techniques such as token sparsification (Rao et al., 2021) and/or quantization (Gholami et al., 2021), and that it applies to newer ViT variants such as EfficientFormer (Li et al., 2022), XCiT (Ali et al., 2021), and Swin (Liu et al., 2021). Moreover, oViT can provide strong results even for pruning highly-accurate CNNs (Wightman et al., 2021; Tan & Le, 2021) and detection models (Carion et al., 2020). In addition, it outperforms all known pruners on the challenging recent SMC-Bench compression benchmark (Anonymous, 2023).

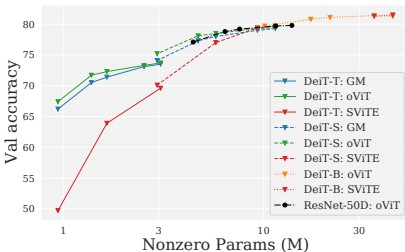

Figure 1: Validation accuracy versus non-zero parameters for DeiT-Tiny, -Small and -Base models, as well as a highly-accurate ResNet50D model, pruned to $\{50\%, 60\%, 75\%, 80\%, 90\%\}$ sparsities using iterative Global Magnitude (GM), SViTE, and oViT methods.

**Overall, our correlation-aware pruner outperforms existing approaches on ViTs, CNNs (ResNet and EfficientNet), and language models (RoBERTa). Specifically for ViTs, our recovery recipes and sparsity sweep framework enable oViT to achieve state-of-the-art compression, roughly doubling sparsity at similar accuracy (Chen et al., 2021), with only half the compression budget.**

**Related work.** Vision Transformers (ViTs) (Dosovitskiy et al., 2020) have set new accuracy benchmarks, but are known to require careful tuning in terms of both augmentation and training hyperparameters. Identifying efficient recipes is an active research topic in itself (Touvron et al., 2021; Steiner et al., 2021). We propose new and general recipes for *fine-tuning* ViTs, which should be useful to the community. Several prior works have investigated ViT compression, but focusing on *structured* pruning, such as removing tokens (Zhu et al., 2021; Kim et al., 2021; Xu et al., 2021; Pan et al., 2021; Song et al., 2022; Rao et al., 2021; Hou & Kung, 2022). We show experimentally that these approaches are *orthogonal* to unstructured pruning, which can be applied in conjunction to further compress these models.

Unstructured pruning, on which we focus here, considers the problem of removing individual network weights, which can be leveraged for computational savings (Kurtz et al., 2020; Hoefler et al., 2021). The only existing prior work on *unstructured* ViT pruning is SViTE (Chen et al., 2021), which applied the general RigL pruning method (Evci et al., 2020) to the special case of ViT models. We also present results relative to well-tuned magnitude pruning, the best existing second-order pruners (Singh &

Alistarh, 2020; Frantar et al., 2021; Kurtic et al., 2022) and AC/DC pruning (Peste et al., 2021). oViT improves upon existing methods across almost all benchmarks, by large margins at high sparsity.

Research on accurate pruning using second-order information was initiated by LeCun et al. (1989); Hassibi et al. (1993), and has recently garnered significant attention, e.g. (Dong et al., 2017; Wang et al., 2019; Singh & Alistarh, 2020; Yu et al., 2022). This approach can lead to good results for both gradual pruning (Frantar et al., 2021; Kurtic et al., 2022) and one-shot (post-training) compression (Frantar & Alistarh, 2022). The closest work to ours is (Frantar et al., 2021; Kurtic et al., 2022), who propose different approximations.

We introduce a new approximation of second-order information for pruning, which can efficiently take weight correlations into account. We show this to be especially-accurate for ViT models. Another source of novelty is our in-depth investigation of sparse fine-tuning approaches for efficient accuracy recovery of ViT models. Together, these two techniques lead to state-of-the-art results.

## 2 BACKGROUND AND PROBLEM SETUP

The pruning problem assumes a fixed model architecture with weights $\mathbf{w} \in \mathbb{R}^d$ ($d$ is the total number of parameters), and aims to find a configuration of weights with as many zeros as possible while preserving the performance of the original dense model. *Gradual* pruning, e.g. (Hoefler et al., 2021), usually starts from an accurate *dense* model, and progressively removes weights by setting them to zero, followed by fine-tuning phases.

**Weight Saliency.** The pruning step usually relies on proxies for weight importance, defined according to certain criteria. For instance, *weight magnitude* is arguably the most popular criterion, e.g. (Han et al., 2015; Zhu & Gupta, 2017; Gale et al., 2019). Specifically, given model $\mathbf{w} \in \mathbb{R}^d$, the saliency of each weight is its absolute value (magnitude) $\rho_j = |w_j|$ for $j \in \{1, 2, \ldots, d\}$; weights with the smallest scores are pruned away. Gradual magnitude pruning is usually a strong baseline across most models and settings. Many other criteria exist, such as gradient magnitude (Evci et al., 2020) or "rates of change" in the weights (Sanh et al., 2020).

**The Optimal Brain Surgeon (OBS).** LeCun et al. (1989) and Hassibi et al. (1993) proposed a framework for obtaining weight saliency scores by leveraging (approximate) second-order information about the loss. Specifically, they start from the Taylor approximation of the loss $\mathcal{L}$ in the vicinity of the dense model parameters $\mathbf{w}^*$. Assuming that $\mathbf{w}^*$ is close to the optimum (hence $\nabla\mathcal{L}(\mathbf{w}^*) \simeq 0$), one seeks a binary mask $\mathbf{M}$ (with elements $\in \{0, 1\}$) and new values for the remaining weights $\mathbf{w}^M$, such that the resulting increase in loss is minimal. A standard approach to approximate the loss increase is to expand the loss function up to the second order in model weights:

$$\mathcal{L}(\mathbf{w}^M) - \mathcal{L}(\mathbf{w}^*) \simeq \frac{1}{2}(\mathbf{w}^M - \mathbf{w}^*)^\top \mathbf{H}_\mathcal{L}(\mathbf{w}^*)(\mathbf{w}^M - \mathbf{w}^*) \tag{1}$$

where $\mathbf{H}_\mathcal{L}(\mathbf{w}^*)$ is the Hessian of the model at $\mathbf{w}^*$, and $\mathbf{w}^M$ represents weights after the pruning step. In this setup, LeCun et al. (1989) and Hassibi et al. (1993) showed that the "optimal" weight to remove, incurring the least loss, and the update to the remaining weights, can be determined via a *closed-form* solution to the above inverse problem. Specifically, the saliency score $\rho_i$ for $i^{\text{th}}$ weight and the optimal weight update $\delta\mathbf{w}$ for the remaining weights after elimination of the $i^{\text{th}}$ weight are as follows:

$$\rho_i = \frac{w_i^2}{2[\mathbf{H}_\mathcal{L}^{-1}(\mathbf{w}^*)]_{ii}}, \quad \delta\mathbf{w}^* = -\frac{w_i}{[\mathbf{H}_\mathcal{L}^{-1}(\mathbf{w}^*)]_{ii}}\mathbf{H}_\mathcal{L}^{-1}(\mathbf{w}^*)\mathbf{e}_i, \tag{2}$$

where $\mathbf{e}_i$ is the $i^{\text{th}}$ basis vector. Theoretically, the procedure would have to be executed one-weight-at-a-time, recomputing the Hessian after each step. In practice, this procedure suffers from a number of practical constraints. The first is that direct Hessian-inverse computation is computationally-infeasible for modern DNNs, due to its quadratic-in-dimension storage and computational costs. This has led to significant recent work on efficient second-order approximations for pruning and quantization (Dong et al., 2017; Wang et al., 2019; Singh & Alistarh, 2020; Yu et al., 2022).

**WoodFisher and the Optimal BERT Surgeon.** The *empirical Fisher* approximation (Amari, 1998) is a classic way of side-stepping some of the above constraints, and can be formally-stated as follows:

$$\mathbf{H}_\mathcal{L}(\mathbf{w}^*) \simeq \mathbf{F}(\mathbf{w}^*) = \lambda\mathbf{I}_{d \times d} + \frac{1}{N}\sum_{i=1}^N \nabla\mathcal{L}_i(\mathbf{w}^*)\nabla\mathcal{L}_i(\mathbf{w}^*)^\top \tag{3}$$

where $\nabla\mathcal{L}_i(\mathbf{w}^*) \in \mathbb{R}^d$ is a gradient computed on a sample of data, $\lambda > 0$ is a dampening constant needed for stability, and $N$ is the total number of gradients used for approximation. Note that the resulting matrix is *positive* definite.

The memory required to store the empirical Fisher matrix is still quadratic in $d$, the number of parameters. Singh & Alistarh (2020) investigated a diagonal block-wise approximation with a predefined block size $B$, which reduces storage cost from $\mathcal{O}(d^2)$ to $\mathcal{O}(Bd)$, and showed that this approach can lead to strong results when pruning CNNs. Kurtic et al. (2022) proposed a formula for block pruning, together with a set of non-trivial optimizations to efficiently compute the block inverse, which allowed them to scale the approach for the first time to large language models.

A second obvious limitation of the OBS framework is that applying the procedure and recomputing the Hessian one weight at a time is prohibitively expensive, so one usually prunes multiple weights at once. Assuming we are searching for the set of weights $Q$ whose removal would lead to minimal loss increase after pruning, we get the following constrained optimization problem:

$$\min_{\delta\mathbf{w}} \frac{1}{2}\delta\mathbf{w}^\top \mathbf{F}(\mathbf{w}^*)\delta\mathbf{w} \quad \text{s.t.} \quad \mathbf{E}_Q\delta\mathbf{w} + \mathbf{E}_Q\mathbf{w}^* = \mathbf{0}, \tag{4}$$

where $\mathbf{E}_Q \in \mathbb{R}^{|Q|\times d}$ is a matrix of basis vectors for each weight in $Q$. The corresponding saliency score for the group of weights $Q$ and the update $\delta\mathbf{w}_{\mathbf{Q}}^*$ of remaining weights is (Kurtic et al., 2022):

$$\rho_Q = \frac{1}{2}\mathbf{w}_Q^{*\top}\left(\mathbf{F}^{-1}(\mathbf{w}^*)_{[Q,Q]}\right)^{-1}\mathbf{w}_Q^*, \quad \delta\mathbf{w}_Q^* = -\mathbf{F}^{-1}(\mathbf{w}^*)\mathbf{E}_Q^\top\left(\mathbf{F}^{-1}(\mathbf{w}^*)_{[Q,Q]}\right)^{-1}\mathbf{w}_Q^*. \tag{5}$$

However, an exhaustive search over all subsets of size $|Q|$ from $d$ elements requires $\binom{d}{|Q|}$ evaluations, which makes it prohibitively expensive for $|Q| > 1$. To alleviate this issue in the case of unstructured pruning Singh & Alistarh (2020) ignores correlations between weights and in the case of block pruning Kurtic et al. (2022) ignores correlations between blocks and solves only for correlations between weights within the same block. Despite these approximations, both approaches yield state-of-the-art results in their respective setups. As we will demonstrate later, our oViT method improves upon these approximations by reformulating this problem and proposing a correlation-aware solution that is fast and memory-efficient even for models as large as DeiT-Base (86M parameters).

## 3 The oViT Pruning Framework

### 3.1 Why Is Pruning Vision Transformers Hard?

The literature suggests that ViT models are difficult to compress: the best known unstructured pruning results for ViT models only achieve 50% sparsity before significant accuracy loss (Chen et al., 2021). It is natural to ask *why*. Our investigation suggests the following factors at play.

**Factor 1: Accurate One-shot Pruning is Hard.** ViT models tend to lose a significant amount of accuracy at each pruning step: Figure 3 shows the accuracy drops of Tiny, Small and Base model variants under the Global Magnitude (GM), WoodFisher/oBERT (WF), and Optimal ViT Surgeon (oViT) pruners, at various sparsities. In this one-shot setup, the higher accuracies of oViT come from the significantly improved sparse projection step. Despite these improvements, accuracy drops are still not negligible, so we need a solid post-pruning recovery phase.

**Factor 2: Accuracy Recovery is Difficult.** Once accuracy has been dropped following a pruning step, it is hard for ViT models to recover accuracy. This is illustrated in Figure 4, showing recovery under gradual pruning for various fine-tuning strategies. Results demonstrate the importance of the learning rate schedule and augmentation recipes.

Our work introduces new techniques to address both these factors: we introduce a new state-of-the-art one-shot pruner for ViTs, complemented with generic recipes for post-pruning fine-tuning.

### 3.2 Ingredient 1: An Efficient Correlation-Aware Second-Order Pruning

Our aim is to solve the pruning problem stated in the previous section: given a weight pruning target $k$, find the optimal set of weights $Q$ to be pruned, such that $|Q| = k$ and the loss increase is minimized. Exactly solving for the optimal $Q$ is an NP-hard problem (Blumensath & Davies, 2008), so we will investigate an iterative greedy method for selecting these weights, similar to the ideal version of the OBS framework discussed above. Importantly, our method *properly considers weight correlations*, while being *fast and space-efficient*. In turn, this leads to significant improvements over other pruners, especially in the context of vision transformers.

Formally, a correlation-aware greedy weight selection approach would perform pruning steps iteratively, as follows. Given a set of already-selected weights $Q$, initially $\emptyset$, we always add to $Q$ the weight $q$ which has minimal joint saliency $\rho_{Q \cup \{q\}}$, repeating until the size of $Q$ equals the pruning target $k$. The fact that we add weights to the set one-by-one allows us to take into account correlations between pruned weights. However, a naive implementation of this scheme, which simply recomputes saliency at each step, would be prohibitively expensive, since it requires $O(kd)$ evaluations of $\rho_Q$, each of which involves an $O(B^3)$ matrix inversion, where $B$ is the Fisher block size.

The centerpiece of our approach is a reformulation of the OBS multi-weight pruning problem in Equation 5 which will allow us to take correlations into account, while being practically-efficient. Specifically, we now show that, when using the empirical Fisher approximation, the problem of finding the optimal set of weights $Q$ to be removed, while taking correlations into account, is equivalent to the problem of finding the set of sparse weights which best preserve the original correlation between the dense weights $\mathbf{w}^*$ and the gradients $\nabla \mathcal{L}_i(\mathbf{w}^*)$ on an fixed set of samples $i \in \mathcal{S}$. Formally, this result, whose proof we provide in Appendix L, is stated as follows.

**Theorem 1.** *Let $\mathcal{S}$ be a set with $m$ samples, and let $\nabla \mathcal{L}_1(\mathbf{w}^*), \ldots, \nabla \mathcal{L}_m(\mathbf{w}^*)$ be a set of their gradients, with corresponding empirical Fisher matrix $\mathbf{F}^{-1}(\mathbf{w}^*)$. Assume a sparsification target of $k$ weights from $\mathbf{w}^*$. Then, a sparse minimizer for the constrained squared error problem*

$$\min_{\mathbf{w}'} \frac{1}{2m} \sum_{i=1}^{m} \left( \nabla \mathcal{L}_i(\mathbf{w}^*)^\top \mathbf{w}' - \nabla \mathcal{L}_i(\mathbf{w}^*)^\top \mathbf{w}^* \right)^2 \text{ s.t. } \mathbf{w}' \text{ has at least } k \text{ zeros}, \quad (6)$$

*is also a solution to the problem of minimizing the Fisher-based group-OBS metric*

$$\underset{Q, |Q|=k}{\arg\min} \frac{1}{2} \mathbf{w}_Q^{*\top} \left( \mathbf{F}^{-1}(\mathbf{w}^*)_{[Q,Q]} \right)^{-1} \mathbf{w}_Q^*. \quad (7)$$

**An Efficient Sparse Regression Solver.** The formulation in Equation (6) reduces pruning to a sparse regression problem, where the "input" is given by *gradients* over calibration samples. A related problem arises in the context of one-shot (post-training) pruning (Hubara et al., 2021; Frantar & Alistarh, 2022), where authors solve a related sparse $\ell_2$-fitting problem, but sparse weights are determined relative to the *layer inputs* rather than the *layer gradients*. Specifically, the OBC solver (Frantar & Alistarh, 2022) observes that, since the loss is quadratic, the row-wise Hessians are independent, and only depend on the layer input. Therefore, the method processes rows independently, and zeroes out weights from each row greedily, one-by-one, in increasing order of squared error. It then *updates* the remaining weights to reduce the $\ell_2$ error. This essentially implements the OBS selection and update in Equation 2 *exactly*, assuming layer-wise $\ell_2$ loss. We build on this strategy to implement our greedy weight-subset selection, with some significant modifications.

A direct implementation of this approach to remove $\Theta(d)$ weights would have $O(d^3)$ runtime, where $d$ is the layer dimension, as the $\Theta(d^2)$-time selection + update process is repeated $\Theta(d)$ times. This can be reduced to $O(d \cdot \max(d, B^2))$ by using the block diagonal structure of the Fisher matrix with block size $B$, when performing updates after each weight elimination. However, as $B$ is typically much smaller than $d$, this would still be extremely slow. We apply a different approach: we treat each block of the Fisher matrix as an independent row for the purposes of the $\ell_2$ solver, and then we merge the all the blocks into a single global approximation of the Fisher matrix. This allows us to perform global weight saliency calculation, and weight updates following Equations 5 and 7. The resulting algorithm, given in full in Appendix 1, requires $O(d \cdot B^2)$ runtime and $O(d \cdot B)$ space. When working with small block sizes, our method is very fast and has practically no overhead over existing Fisher-based OBS approaches, while yielding significantly improved one-shot pruning results. Detailed pseudo-code and an efficient implementation are provided as supplementary material.

## 3.3 INGREDIENT 2: FINE-TUNING AND PRUNING PROCEDURE

ViTs are notoriously difficult to train: they need longer training relative to CNN architectures, and the choice of the training procedure (learning rate schedule, regularization, augmentation) can have a major impact on convergence and accuracy (Touvron et al., 2021; Steiner et al., 2021). We found the same to apply to post-pruning accuracy recovery, which is key in gradual pruning; below, we describe the main ingredients to obtaining highly-accurate fine-tuning schedules as part of our method.

**Learning Rate Schedule.** First, to achieve good performance during gradual pruning, the learning rate (LR) schedule is crucial. Specifically, we propose to use a *cyclic linear* schedule:

$$\eta(t) = \eta_{\max} - (\eta_{\max} - \eta_{\min})\frac{t\%T}{T}, \tag{8}$$

where $\%x$ means taking the remainder after integer division by $x$. We chose a linear decay for simplicity; we obtained similar results for other functions (e.g., cubic decay). By contrast, as we illustrate in Section 4.2, the *cyclic* nature of the schedule is key for accurate pruning. Specifically, this choice is justified theoretically by tying back to the original assumptions of the OBS framework: for Equation 1 to hold, the pruned model should be well-optimized (i.e. have small gradients) at the point when pruning is performed. Moreover, right after the pruning step, having a larger value of the learning rate is useful since it gives the model a chance to recover from the sub-optimal point induced via pruning. We note that this learning rate schedule is different from prior work on pruning, which typically uses a single decay cycle (Kusupati et al., 2020; Singh & Alistarh, 2020; Peste et al., 2021), or dynamic learning rate rewinding, e.g. (Frankle et al., 2019; Renda et al., 2020).

**Regularization and Augmentation.** Another important ingredient for achieving high accuracy is the choice of the regularization/augmentation pipeline. Specifically, we observe that smaller models such as DeiT-Tiny benefit from *lower* levels of data augmentation during fine-tuning as in Steiner et al. (2021), whereas larger models such as DeiT-Base behave best with more complex augmentation and regularization, such as Touvron et al. (2021). Intuitively, fine-tuning sparsified small models with high augmentation likely may exceed model capacity, rendering the optimization process unstable. We provide detailed parameter values and ablations for this training component in the AppendixB.

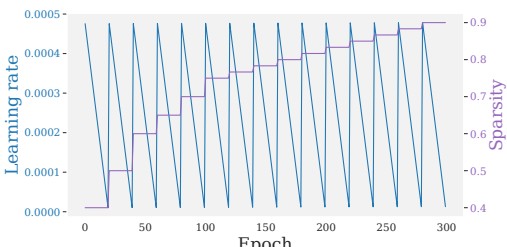

Figure 2: Blue: Cyclic linear learning rate schedule used in the work. Violet: Dependence of the global model sparsity on the epoch. Every change in sparsity corresponds to a pruning step.

**An Efficient Pipeline for Sparsity Sweeps.** We propose a simple iterative pruning framework, which takes a set of target sparsity configurations and produces models which match these configurations *in a single run*. Specifically, we start from a standard gradual pruning setup, which prunes in a sequence of steps of increasing sparsity, followed by sparse fine-tuning. We then set the intermediate values in such a way that all intermediate target sparsity levels are achieved. For example, if one wishes to obtain checkpoints with sparsity levels $40\%, 50\%, 75\%, 90\%$, one can set the lowest sparsity level on the gradual pruning schedule to $40\%$, the highest sparsity level to $90\%$ ,and $50\%, 75\%$ as intermediate points. Between any two such pruning steps, we apply the cyclic retraining schedule above.

We emphasize the fact that *having an accurate pruner is key* to support this compressed pruning approach: virtually all previous high-accuracy pruning methods, e.g. (Kusupati et al., 2020; Singh & Alistarh, 2020) redo the *entire training run* for each sparsity target in turn. In our experimental section, we also examine the impact of additional fine-tuning applied to each checkpoint, and show that it induces small-but-consistent improvements.

## 4    EXPERIMENTAL SETUP AND RESULTS

**Setup and Goals.** We consider the ImageNet (Russakovsky et al., 2015) image classification benchmark, and aim to examine how sparsity impacts accuracy for different model variants. We consider three scenarios: *one-shot, single-step pruning* of a pretrained model, where performance is clearly tied to the quality of the second-order approximation, *one-shot + fine-tuning*, in which we follow one-shot pruning by a short period of fine-tuning, and, finally, *iterative gradual pruning*, where one applies pruning periodically, with some retraining interval, gradually increasing sparsity.

### 4.1    ONE-SHOT PRUNING, WITHOUT AND WITH FINE-TUNING

We start by examining the quality of existing one-shot pruners relative to oViT. We compare against carefully-tuned variants of Magnitude Pruning (Magn), Gradient times Weight (GrW) Sanh et al. (2020), and the SOTA second-order methods WoodFisher (Singh & Alistarh, 2020; Kurtic et al., 2022) and M-FAC (Frantar et al., 2021). Our tuning process, optimal hyper-parameter choices, and ablations are detailed in Appendices G and H. To demonstrate the importance of weight correlations,

we also compare with an implementation of LeCun et al. (1989) via WoodFisher, denoted by WF-1 on the Figure 3, which performs diagonal approximation.

For Magnitude and oViT we present results for both *uniform* layer-wise sparsity and *global* sparsity. This choice does not make a difference for Magnitude, whereas global sparsity is better for oViT, as it can adjust saliency globally. We only investigate global sparsity for the other methods, in Figure 3.

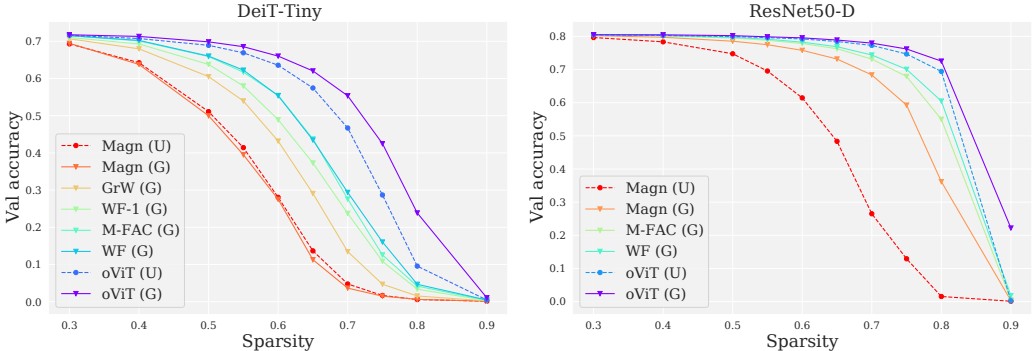

Figure 3: One-shot pruning for DeiT-Tiny (**left**) and ResNet50-D (**right**).

The full comparison between all these SOTA methods is presented for DeiT-Tiny and a highly-accurate ResNet50-D, in Figure 3. First, all first and second-order methods outperform Magnitude. All 2nd order methods are better than the 1st order saliency described above. WoodFisher with small block size is better than both the diagonal approximation and large-block M-FAC. This suggests that it is beneficial to take weight correlations into account, but attempting to incorporate dependencies between large groups of weights may lead to noisy estimates that are detrimental to performance. We also present comparison between different pruners on several variants of ViT models in Appendix C.

**We note that oViT outperforms all other methods, on both DeiT-Tiny and ResNet50, setting new SOTA results.** Remarkably, the margin is so large that *oViT with uniform sparsity* still outperforms *global sparsity* WoodFisher. The computational cost of WF is approximately the same as for oViT: oViT pruning step on DeiT-Small takes 23 minutes, compared to 20 minutes for WF. (The majority of the cost for both methods comes from the collection of gradients, not Hessian estimation.)

**Oneshot + Fine-tuning.** Next, we consider the one-shot + fine-tune setup where one first prunes the model to a certain sparsity level, and then performs a short period of fine-tuning. Specifically, below we prune to $50\%$ sparsity, and fine-tune for 20 epochs. In addition to ViT/DeiT, we also consider similar models based on variants of self-attention (Liu et al., 2021; Ali et al., 2021), and compare against a simple GM baseline. We use a linearly-decaying learning rate schedule between $\eta_{max} = 10^{-4}$ to $\eta_{max} = 10^{-5}$ and the training DeiT training recipe (Touvron et al., 2021). The results are given in Table 1, and show that oViT can almost fully-recover accuracy in this setup for all models; the gaps from GM and WF (see DeiT-Small 75 and 90%) are still very significant. Given this, in the following experiments, we will mainly focus on oViT as the second-order pruner.

Table 1: One-shot + fine-tuning on ImageNet.

| Model | Method | Sparsity (%) | Top1-Accuracy (%) |
|---|---|---|---|
| | Dense | 0 | 79.8 |
| | GM | 50 | 79.0 |
| | oViT | | **79.5** |
| DeiT-Small | GM | 75 | 74.3 |
| | WF | | 75.8 |
| | oViT | | **76.9** |
| | GM | 90 | 45.6 |
| | WF | | 59.3 |
| | oViT | | **65.1** |
| | Dense | 0 | 81.8 |
| | GM | 50 | 81.5 |
| | oViT | | **81.6** |
| DeiT-Base | GM | 75 | 80.1 |
| | WF | | 80.2 |
| | oViT | | **81.0** |
| | GM | 90 | 68.1 |
| | WF | | 69.2 |
| | oViT | | **76.3** |
| | Dense | 0 | 82.0 |
| XCiT-Small | GM | 50 | 81.7 |
| | oViT | | **81.9** |
| | Dense | 0 | 81.3 |
| Swin-Tiny | GM | 50 | 80.6 |
| | oViT | | **80.9** |

## 4.2 POST-PRUNING RECOVERY

For ViTs, the choice of augmentation parameters and learning rate schedule is critical. For example, reducing the level of augmentation during fine-tuning for smaller models, e.g. DeiT-Tiny, significant improves performance, whereas larger models, e.g. the 4x larger DeiT-Small, requires

strong augmentations for best results even during fine-tuning. See Figure 4 for an illustration; we provide full details and results in the Appendix B.

Moreover, the choice of cyclic learning rate (LR) schedule is critical as well. To illustrate this, we compare convergence obtained when using a *cosine annealing* schedule, which is very popular for pruning CNNs (Kusupati et al., 2020; Singh & Alistarh, 2020; Peste et al., 2021), from $\eta_{max} = 5 \cdot 10^{-4}$ to $\eta_{min} = 10^{-5}$, while performing pruning updates 2 times more frequently (one update per 10 epochs) than in our standard setup from the following section 4.3. The results are provided in Figure 4, where cosine annealing (no cycles) is in red. All experiments use the oViT pruner, and highlight the importance of the learning rate and augmentation schedules for recovery.

Table 2: Accuracy-vs-sparsity for gradual pruning on ImageNet. (↑) denotes accuracy after additional fine-tuning for 100 epochs.

| Model | Method | Sparsity (%) | FLOP Reduction (%) | Top1 Accuracy (%) |
|---|---|---|---|---|
| | Dense | 0 | 0 | 72.2 |
| | GM | | 43.9 | 73.5 |
| | oViT | 50 | 43.9 | **73.7** |
| | SViTE-Tiny | | 43.9 | 69.6 |
| | GM | 60 | 52.6 | 73.1 (73.2 ↑) |
| | oViT | | 52.6 | 73.3 (**73.6** ↑) |
| DeiT-Tiny | GM | | 65.8 | 71.4 (71.9 ↑) |
| | oViT | 75 | 65.8 | **72.3** (72.6 ↑) |
| | SViTE-Tiny | | 65.8 | 63.9 |
| | GM | 80 | 69.7 | 70.5 (70.9 ↑) |
| | oViT | | 70.2 | 71.7 (**72.0** ↑) |
| | GM | | 79.0 | 66.2 (66.6 ↑) |
| | oViT | 90 | 79.0 | 67.4 (**68.0** ↑) |
| | SViTE-Tiny | | 79.0 | 49.7 |
| | Dense | 0 | 0 | 79.8 |
| | GM | | 46.7 | 79.3 (79.8 ↑) |
| | oViT | 50 | 46.9 | 79.4 (**79.9** ↑) |
| | SViTE-Small | | 46.3 | 79.7 |
| | GM | | 56.1 | 79.0 (79.5 ↑) |
| | oViT | 60 | 56.2 | 79.3 (**79.8** ↑) |
| DeiT-Small | SViTE-Small | | 55.4 | 79.4 |
| | GM | | 70.1 | 78.0 (78.7 ↑)) |
| | oViT | 75 | 70.2 | 78.5 (**79.0** ↑) |
| | SViTE-Small | | 70.3 | 77.0 |
| | GM | 80 | 74.2 | 77.3 (77.9 ↑) |
| | oViT | | 74.9 | 78.0 (**78.6** ↑) |
| | GM | | 84.0 | 74.1 (74.7 ↑) |
| | oViT | 90 | 84.1 | 75.2 (**75.8** ↑) |
| | SViTE-Small | | 84.1 | 70.1 |
| | Dense | 0 | 0 | 81.8 |
| | oViT | 50 | 48.5 | **81.6** |
| | SViTE-Base | | 48.0 | 81.5 |
| DeiT-Base | oViT | 60 | 58.2 | **81.5** |
| | SViTE-Base | | 57.5 | 81.3 |
| | | 75 | 72.8 | 81.1 (81.2 ↑) |
| | oViT | 80 | 77.7 | 80.8 (81.1 ↑) |
| | | 90 | 87.4 | 79.7 (80.1 ↑) |

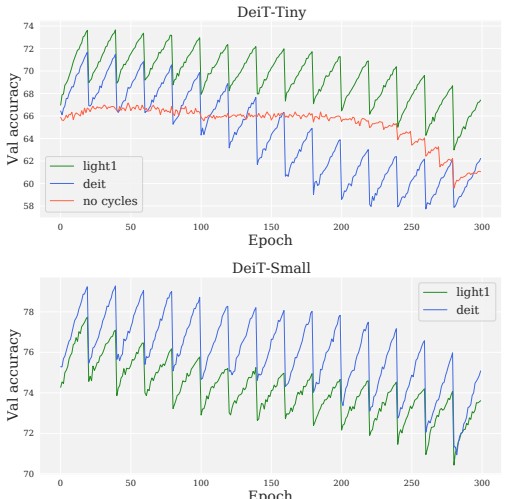

Figure 4: Ablations of the training setting on DeiT-Tiny (up) and DeiT-Small (down). Green curves correspond to the *light1* (Steiner et al., 2021) augmentation recipe, blue curves to the *deit* (Touvron et al., 2021) recipe. The red curve follows training with a single (acyclic) cosine annealing schedule, as in (Kusupati et al., 2020; Singh & Alistarh, 2020).

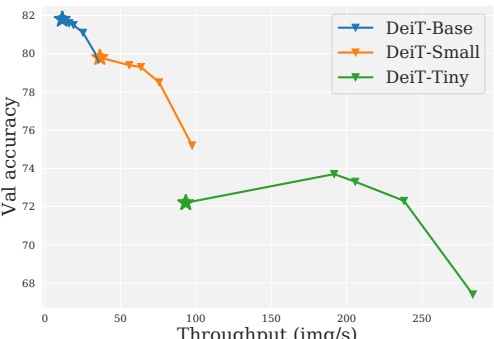

Figure 5: Accuracy vs. throughput for dense and sparse ViT models when executing on the DeepSparse inference engine. ⋆ corresponds to dense models.

## 4.3 GRADUAL PRUNING RESULTS

Finally, we execute gradual pruning with the sparsity schedule, augmentation choices, and cyclic linear learning-rate scheduler discussed above. The whole gradual pruning procedure lasts for 300 epochs, as in Touvron et al. (2021). We aim to obtain accurate sparse checkpoints for 50%, 60%, 75%, 80%, and 90% sparsity. For this, we prune to 40% in the initial step, and increment sparsity every 20 epochs, until reaching 90%, with fine-tuning in between. (See Figure 4.) We select the

accuracy of intermediate models which match the target sparsities; to examine the impact of fine-tuning, we trained each of the resulting sparse checkpoints for an additional 100 epochs, marked with (↑). We compare with global magnitude (GM) following the same schedule as oViT, as well as the state-of-the-art SViTE (Chen et al., 2021) paper, which trains the sparse model from scratch using a variant of RigL (Evci et al., 2020), but for a total of 600 epochs. The results are in Table 2.

For DeiT-Tiny and 50% sparsity, we achieve significant improvements upon SViTe, and even manage to improve test accuracy relative to the dense model. We believe this is due to more proper choice of augmentation for fine-tuning this smaller model. At 75-80%, we recover the dense model accuracy, showing for the first time that ViT models can be pruned to such sparsities without loss. We observe a significant accuracy drop only at 90%. GM pruning also benefits from the choices made in our schedule, outperforming SViTe at 50% sparsity; yet, there are significant gaps in favor of oViT at higher sparsities, as expected.

On the 4x larger DeiT-Small model, SViTE performs remarkably well at 50% sparsity (79.7%), almost matching the dense model, but oViT outperforms it very slightly after fine-tuning (79.9%). In terms of total training budget, SViTE uses 600 epochs to produce each model (and so, the 50%-sparse one as well), whereas we use a total of 40 epochs for pruning to 50% + initial fine-tuning, and 100 additional epochs for sparse model fine-tuning. Even if we take into account the original 300 epochs for training the publicly-available dense DeiT checkpoint (Touvron et al., 2021), our approach is significantly more efficient (440 vs. 600 epochs), and savings compound across sparse models. At 75% sparsity, oViT drops $\sim 1\%$ of accuracy relative to dense post-finetuning, with a significant gap of $1\%$ Top-1 relative to GM, and $2\%$ Top-1 relative to SViTE. The trend continues for higher sparsities, where we note a remarkable gap of $5.7\%$ Top-1 vs SViTE at 90% sparsity. We obtain similar numbers for DeiT-Base; generally, we achieve $\geq 99\%$ recovery at $\geq 75\%$ sparsity.

**Additional Experiments.** Due to space constraints, several experiments are deferred to the Appendix. Specifically, we show that oViT also provides high accuracy when pruning other highly-accurate models (EfficientFormer, ResNet50D, and EfficientNetV2) in Appendix D, while having similar runtime to the fastest implementation of WoodFisher (Kurtic et al., 2022) (Appendix E). In Appendix F we show that oViT can also be applied in conjunction with other types of compression, in particular token pruning (Rao et al., 2021), quantization-aware training, and semi-structured (2:4) sparsity. Specifically, we show that oViT can induce 2:4 sparsity with minor accuracy drop across DeiT models. In Appendix J, we show oViT also outperforms AC/DC pruning (Peste et al., 2021), while in Appendix M we show that oViT outperforms all other methods on SMC-Bench (Anonymous, 2023).

**Sparse Speedups.** Finally, in Figure 5, we examine the speedups obtained by oViT from unstructured sparsity, when executed on a sparsity-aware CPU inference engine (Kurtz et al., 2020). Specifically, we executed the models from Table 2 using 4 cores of an Intel(R) Xeon(R) Gold 6238R CPU, at batch size 64. We find it interesting that sparse ViTs build an almost-contiguous Pareto frontier from 82% to 68% Top-1 accuracy (Y axis), with a 25x span in throughput (from 11 imgs/second to 260 imgs/second, X axis). Notably, the DeiT-Tiny model obtains a speedup of 2.4x without any accuracy loss (thanks to the accuracy increase), while Base and Small ViTs show 1.5x speedups with minimal loss of accuracy. Thus, sparsity can be a promising approach to speeding up ViTs. We note that competing methods (e.g. SViTE) would provide similar speedups at the same sparsity levels, but lower accuracy, as shown in Table 2.

## 5 DISCUSSION, LIMITATIONS AND FUTURE WORK

We examined the trade-off between parametrization and accuracy in the context of unstructured pruned ViT models, and presented a new pruner called oViT, based on a new approximation of second-order information, which sets a new state-of-the-art sparsity-accuracy trade-off. Specifically, we have shown for the first time that ViT variants can support significant weight pruning ($\geq 75\%$) at relatively minor accuracy loss ($\leq 1\%$), similar to CNNs. Therefore, our results seem to suggest that, despite their weaker encoded inductive biases, ViT models do not require over-parametrization post-training, and in fact can be competitive with CNNs in terms of accuracy-per-parameter. Our approach extends to other model families, and is complementary to other compression approaches.

One limitation is that we still require rather high computational budgets for best results. This is currently an inherent limitation of ViTs; to address it, we provide efficient oneshot + fine-tune recipes, leading to good results. Further, we have only focused on Top-1 as a leading accuracy metric, without covering transfer accuracy or potential bias (Hooker et al., 2020). Future work should also be able to extend our pruner to *structured* compression of ViT models, or employ our oViT pruner inside different, more computationally-intensive pruning algorithms such as RigL (Evci et al., 2020).

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

## A  OVIT ALGORITHM DESCRIPTION

The section below illustrates in the oViT pruning algorithm step-by-step. Prunable model weights $\mathbb{R}^d$ are partitioned into blocks of fixed size $B$. Below $\rho_i^{(B)}$ denotes the saliency scores for weight $i^{\text{th}}$ inside a block it belongs to and $\rho_i$ is the score across the whole model. The steps of the algorithm are listed below:

---

**Algorithm 1** oViT pruning algorithm

---

 1: $\rho_i$ - saliency scores for weights
 2: Accumulate Fisher inverse blocks $\mathbf{F}$
 3: **for** each block **do**
 4:     err = 0
 5:     **for** element in a block **do**
 6:         Select the weight $w_i$ with smallest score $\rho_i^{(B)}$ (using the equation 2 for $\rho_i$)
 7:         Prune $w_i$
 8:         Update remaining weights in the block via equation 2
 9:         err+ $= \rho_i^{(B)}$
10:         $\rho_i \leftarrow$ err
11:         Save current state of the block for later merging
12:         Update Fisher inverse block
13:     **end for**
14: **end for**
15: Sort the scores $\rho_i$ in ascending order
16: Mark the weights with smallest scores $\rho_i$ as pruned
17: **for** each block **do**
18:     Load the saved state of the block with the weights marked pruned and all remaining alive.
19: **end for**

---

## B  TRAINING DETAILS

**Augmentation/regularization recipe**

Table 3: Summary of the augmentation and regularization procedures used in the work.

| Procedure | DeiT | light1 |
|---|---|---|
| Weight decay | 0.05 | 0.03 |
| Label smoothing $\varepsilon$ | 0.1 | 0.1 |
| Dropout | ✗ | ✗ |
| Stoch.Depth | 0.1 | 0.0 |
| Gradient Clip. | ✗ | 1.0 |
| H.flip | ✓ | ✓ |
| RRC | ✓ | ✓ |
| Rand Augment | 9/0.5 | 2/0.5 |
| Mixup alpha | 0.8 | 0.0 |
| Cutmix alpha | 1.0 | 0.0 |
| Erasing prob. | 0.25 | 0.0 |
| Erasing count | 1 | 0 |
| Test crop ratio | 0.9 | 0.9 |

For the gradual pruning experiments (with 300 epochs) we have used cyclic learning schedule, with high learning rate directly after the pruning step with gradual decrease up to the next pruning step.

Table 4: Hyperparameters of the schedules used in gradual pruning.

| Model | Prune freq | LR sched $\{f_{\text{decay}}, \eta_{\max}, \eta_{\min}\}$ | Augm | Batch size | Epochs |
|-------|-----------|----------------------------------------------------------|------|-----------|--------|
| DeiT-Tiny | 20 | $\{\text{cyclic\_linear}, 5 \cdot 10^{-4}, 1 \cdot 10^{-5}\}$ | *light1* | 1024 | 300 |
| DeiT-Small | 20 | $\{\text{cyclic\_linear}, 5 \cdot 10^{-4}, 1 \cdot 10^{-5}\}$ | *deit* | 1024 | 300 |

For both DeiT-Tiny and DeiT-Small model during the additional fine-tuning for 100 epochs we've applied cosine annealing schedule with $\eta_{\max} = 5 \cdot 10^{-5}, \eta_{\min} = 1 \cdot 10^{-5}$ and all other parameters the same as in the Table 4.

## C  ONE SHOT PRUNING OF DIFFERENT VIT VERSIONS.

In this section we present comparison of Global Magnitude (GM), WoodFisher (WF) and oViT pruners in one-shot pruning setting for DeiT models (Touvron et al., 2021) of diffent size (i.e DeiT-Tiny, DeiT-Small, DeiT-Base) to study the scaling behaviour of ViT sparsification.

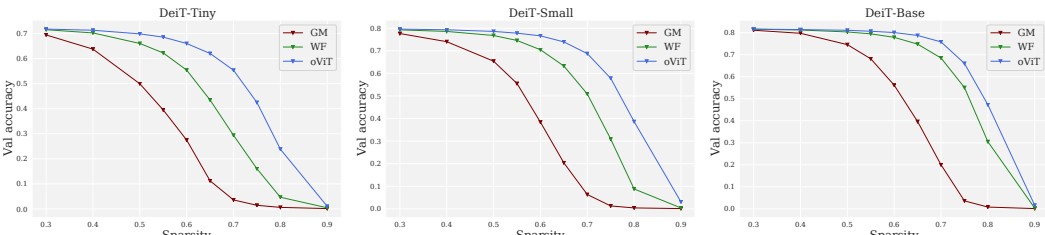

Figure 6: One-shot pruning of different DeiT versions.

Notice, that the gap between magnitude pruning and second order methods is very pronounced for all models, whereas the difference in performance between oViT and WF decreases with increase of the size of model. Nevertheless, oViT performs still noticeably better than WF, especially in high sparsity regime.

## D  EXPERIMENTS WITH OTHER MODELS

Despite the fact that in this work we aim at the pruning of ViT models the proposed approach can be applied to any architecture for image classification, in particular, convolutional neural network (CNN) or a ViT-CNN Hybrid. We have applied oViT gradual pruning to the recently proposed EfficientFormer (Li et al., 2022) using the same setting and hyperparameters as for DeiT-Small. Two CNN architectures - ResNet50-D [1] and EfficientNetV2-Tiny (Tan & Le, 2021) [2], considered in this work were trained with the use of augmentation and regularization procedure described in the recent PyTorch blog post. Differently from most of the prior art we have used the ResNet50-D trained with the modern recipe from timm repository.

For ResNet50-D we prune all convolutional weights except of the first convolution and we keep the classification layer dense. In EfficientNetv2-Tiny we do not prune depthwise convolutions since each channel is processed separately and the only block size that makes sense in this case would be at most of the size of the convolutional kernel spatial dimensions product. We have set the block size to be 256 for ResNet50-D and 16 for EfficientNetV2-Tiny while keeping all the other hyperparameters of oViT the same as for DeiT experiments. Such a small block size was chosen for EfficientNetV2-Tiny due the fact that it is the largest common divisor of the prunable weights.

First of all, we conducted comparison between one-shot pruning methods for ResNet50-D. We compare between Uniform and Global magnitude pruning, WoodFisher with block size of 256, M-FAC with block size of 2048 and oViT with uniform and global sparsity. One can observe that oViT outperforms all previous methods even when comparing uniform sparsity with global sparsity.

---

[1] `resnet50d` checkpoint with 80.5 % accuracy for dense model

[2] `efficientnetv2_rw_t` checkpoint with 82.3 % accuracy for dense model

Contrary to the case of DeiT where there is no much difference in performance between uniform and global magnitude pruning for ResNet50-D global sparsity turns out to be much better. This results is quite expectable since CNN are not uniform and deeper layers are mode wide than those close to the input.

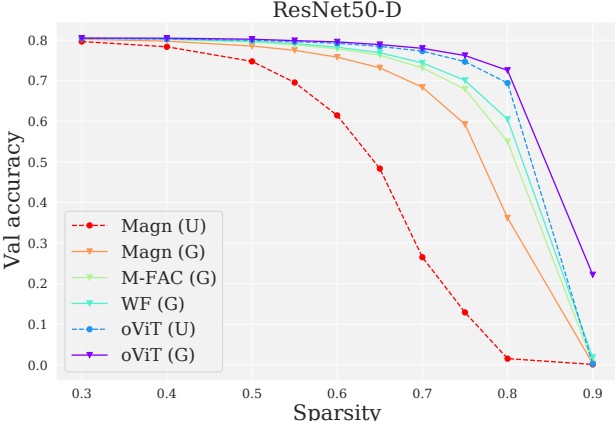

Figure 7: Comparison between different pruning methods on pretrained ResNet50-D model on the ImageNet dataset in one-shot pruning setup.

Next we carried out one-shot + finetuning experiments with ResNet50-D keeping setup the same as for DeiT models in Table 1. We have selected 75 % and 90 % as the difference between methods becomes pronounced only at sufficiently high sparsity. Notably, the performance of Global magnitude and WF is roughly the same, the initial difference after one-shot pruning between WF and Global Magnitude vanishes during the finetuning procedure, whereas there is still a gap in performance between oViT and other methods.

Table 5: One-shot + fine-tuning on ImageNet.

| Model | Method | Sparsity (%) | Top1-Accuracy (%) |
|---|---|---|---|
| | Dense | 0 | 80.5 |
| | GM | | 79.0 |
| | WF | 75 | 79.0 |
| ResNet-50D | oViT | | **79.2** |
| | GM | | 74.7 |
| | WF | 90 | 74.8 |
| | oViT | | **75.2** |

Finally we conducted gradual pruning runs following the same sparsification schedule as for DeiT-models. The EfficientFormer and EfficientNet models despite being already very optimized and parameter efficient can be still compressed with small drop in accuracy.

## E    TIMINGS

Any algorithms involving second order loss information are believed to require tremendous amounts of compute. Time required for calculation of pruning scores and the OBS update comprises collection of grads, Fisher inverse rank-1 updates and additional pruning iteration for oViT. We have measured the time of single pruning step for DeiT-Small and present the results in Table 7. All measurements were done on a single RTX A6000 GPU with 48GB of memory. One can observe that the amount of time needed to perform a pruning update is not very large, especially when compared to the duration of typical training procedure of modern computer vision models on ImageNet that usually takes several days on a multi-gpu node. Note that the additional step for oViT adds small fraction of total computational time relative to other steps of the OBS method.

Table 6: Performance of gradual pruning on ImageNet. Numbers in the parentheses followed by the upwards directed arrow denote additional fine-tuning for 100 epochs.

| Model | Method | Sparsity (%) | Top1-Accuracy (%) |
|---|---|---|---|
| EffFormer-L1 | Dense | 0 | 78.9 |
| | oViT | 50 | 78.0 |
| | | 60 | 77.4 |
| | | 75 | 76.4 |
| | | 90 | 72.4 (72.8 ↑) |
| ResNet-50D | Dense | 0 | 80.5 |
| | oViT | 50 | 79.8 |
| | | 60 | 79.7 |
| | | 75 | 79.2 (79.6 ↑) |
| | | 90 | 77.1 (77.5 ↑) |
| EffNetV2-Tiny | Dense | 0 | 82.4 |
| | oViT | 50 | 81.0 |
| | | 60 | 80.6 |
| | | 75 | 79.6 (80.0 ↑) |
| | | 90 | 75.0 |

Table 7: Minutes per pruning step for DeiT-Small.

| Model | Method | Time (minutes) |
|---|---|---|
| DeiT-Small | Fast WoodFisher (Kurtic et al., 2022) | 20 |
| | oViT | 23 |

## F  COMPOSITE COMPRESSION

In addition to weight pruning one can decrease storage and inference cost with the help of other compression approaches: quantization (casting weights and activations to lower precision) and token pruning specific for the transformer architecture.

### F.1  QUANTIZATION-AWARE TRAINING

Weight quantization is done in the following way - one takes sparse checkpoint and then runs quantization aware training (QAT). We ran QAT training for 50 epochs with linearly decaying learning rate schedule from $\eta_{max} = 10^{-4}$ to $\eta_{min} = 10^{-5}$. Models are quantized to 8-bit precision. In all experiments performed accuracy of quantized model almost reproduces the accuracy of the sparse model stored in full precision.

Table 8: ImageNet-1K top-1 accuracy for sparse models after QAT training.

| Model | Sparsity (%) | Accuracy (%) |
|---|---|---|
| DeiT-Tiny | 75 | 72.2 |
| DeiT-Small | 75 | 77.7 |
| DeiT-Base | 75 | 81 |

### F.2  TOKEN PRUNING

There are different approaches for token pruning proposed in the literature. In this work we follow the one from (Rao et al., 2021). Specifically, in DynamicViT one selects the ratio of tokens being pruned at each step with the lowest importance score, predicted by the model itself. Following the main setup from the paper we prune tokens after $3^{rd}$, $6^{th}$, $9^{th}$ block, and the token pruning ratio after each block is $\rho = 0.2$ (i.e 20% least improtant tokens are pruned).

Table 9: ImageNet-1K top-1 accuracy for sparse models with token pruning.

| Model | Method | Sparsity (%) | Top1-Accuracy (%) |
|---|---|---|---|
| DynamicViT-Tiny | oViT | 50 | 72.0 |
| | | 60 | 71.6 |
| | | 75 | 70.2 |
| DynamicViT-Small | oViT | 50 | 79.5 |
| | | 60 | 79.4 |
| | | 75 | 78.7 |

### F.3 SEMI-STRUCTURED SPARSITY.

While CPUs can utilize sparsity patterns of arbitrary form to speed-up the computations at the present time modern GPU accelerators can handle only restricted form of unstructured sparsity, namely the $N : M$ sparsity pattern that enforces exactly $N$ non-zero values for each block of $M$ weights. Namely, since the introduction of Ampere architecture NVIDIA GPUs have special kernels that can work with $2 : 4$ sparse matrices (Mishra et al., 2021). One can integrate the $N : M$ sparsity in the oViT framework without significant changes. The only difference with the original oViT approach is that while running the oViT iterations one doesn't prune a given weight in case in a group of $M$ weights to which this weights belongs to there are $M - N$ zero weights. Since the sparsity pattern is significantly constrained compared to generic unstructured sparsity pattern drop in performance after doing pruning step and consequent fine-tuning is more challenging than it would be for unconstrained sparsity. In experiments below we prune models to $2 : 4$ sparsity and fine-tune them for 50 epochs with linearly decaying learning rate schedule.

Table 10: Semi-structured $2 : 4$ pruning of ViT models.

| Model | Accuracy (%) |
|---|---|
| DeiT-Tiny | 72.7 |
| DeiT-Small | 79.0 |
| DeiT-Base | 80.5 |

## G OVIT/WF HYPERPARAMETERS

Following the oBERT's directions (Kurtic et al., 2022) on identifying the optimal set of hyperparameters via one-shot pruning experiments, we conduct a grid search over the three most important hyperparameters:

- Number of grads collected for Fisher inverse
- Dampening constant $\lambda$
- Block size

The more grads are collected, the more accurate is the empirical Fisher inverse estimate, however, more compute is required at the same time. We chose $N = 4096$ as a point from which further increase of Fisher samples doesn't improve performance a lot. Dependence of the one-shot pruning performance at different sparsities vs number of grads is presented on Figure 8.

The next parameter to be studied is the dampening constant $\lambda$ in. This constant regularizes the empirical Fisher matrix and allows to avoid instabilities in computation of the inverse. However, this constant decreases the correlation between different weights and in the limit $\lambda \to \infty$ OBS reduces to magnitude pruning. The optimal dampening constant for oViT ($\lambda_{\mathrm{opt}} = 10^{-8}$) is smaller than the one for WoodFisher ($\lambda_{\mathrm{opt}} = 10^{-6}$), i.e oViT remains numerically and computationally stable with smaller amount of regularization compared to WF (we observed that for $\lambda < 10^{-7}$ WF performance starts to deteriorate rapidly).

And the last but not the least important parameter is the block size in (Singh & Alistarh, 2020). The larger the block size is, the more correlations between different weights are taken into account. However, as mentioned in 2 the computational and storage cost scales with the block size. Moreover,

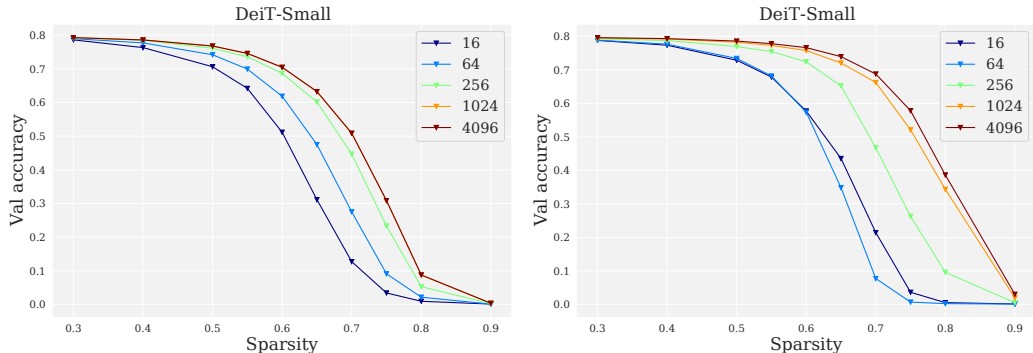

Figure 8: **Left**: One-shot pruning performance of WoodFisher. **Right**: One-shot pruning performance of oViT.

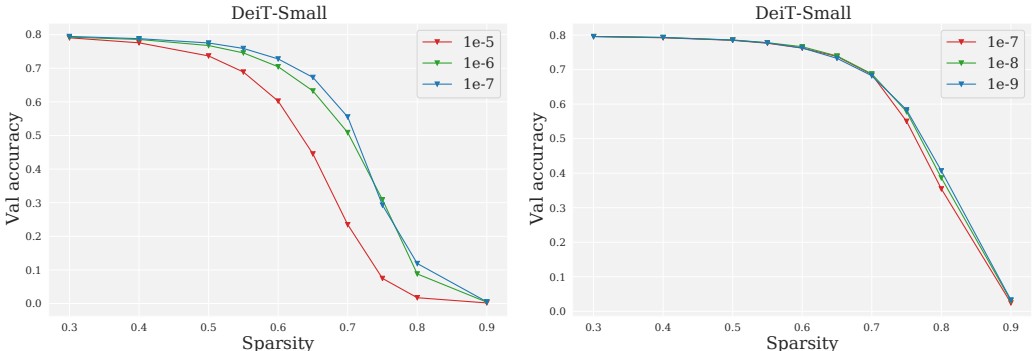

Figure 9: **Left**: One-shot pruning performance of WoodFisher. **Right**: One-shot pruning performance of oViT.

for a fixed number of gradients in the Fisher estimate matrix with larger block sizes is likely to be worse conditioned. Therefore, one would like to work with smaller block sizes but not to keep the approximation as accurate as possible. We've selected block size according to the accuracy-efficiency trade-off.

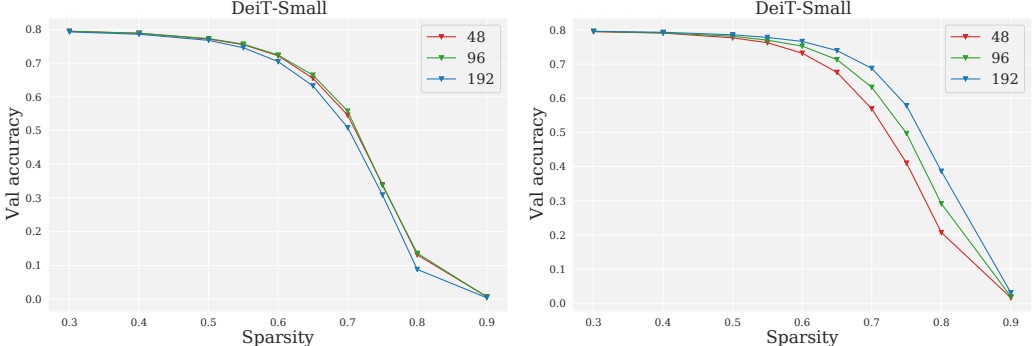

Figure 10: **Left**: One-shot pruning performance of WoodFisher. **Right**: One-shot pruning performance of oViT.

In addition, we've studied the benefit from application of multiple recomputations in the one-shot pruning setting for WoodFisher and oViT. Since the assumption of static Fisher matrix $\mathbf{F}(\mathbf{w}^*)$ doesn't hold in general, we expect that multiple recomputations are likely to result in higher one-shot accuracy in accordance with the result from (Frantar et al., 2021). This is indeed the case. The gain from recomputations is more pronounced for WoodFisher, since oViT already performs implicit Fisher inverse updates in its operation. Yet, the efect is not vanishing even for the case of oViT.

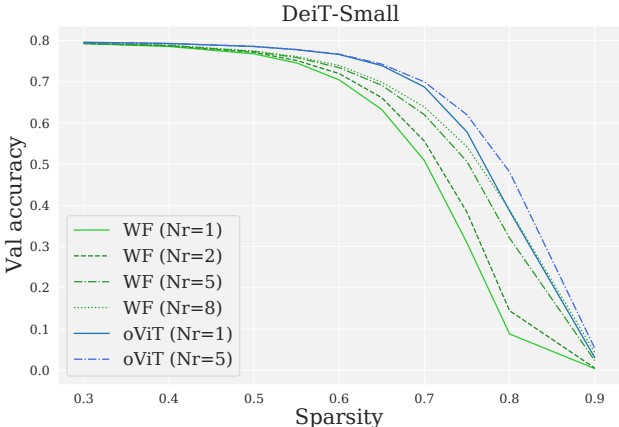

Figure 11: One-shot performance for WF and oViT with different number of recomputations $N_r$.

## H  DETAILS AND HYPERPARAMETER CHOICES FOR OTHER PRUNING METHODS.

In this section we provide some additional details about methods compared on Figures 3 and 7.

The popular Movement Pruning (Sanh et al., 2020) computes the weight saliency scores during the training procedure, hence it is not a one-shot pruning method by the definition. We have observed that use of the naive elementwise product of gradient and weights ( i.e $\rho_i = w_i \odot \nabla_{w_i} \mathcal{L}(w)$) leads to a poor performance, significantly below even the Magnitude Pruning baseline. However, the following first order pruning criterion:

$$\rho_i = \sum_{k=1}^{N} \|w_i^{(k)} \odot \nabla_{w_i} \mathcal{L}^{(k)}(w)\| \tag{9}$$

allows to get reasonable saliency scores that produce more accurate sparse models than Magnitude Pruning. However, its performance is still inferior to any of the second order pruning methods. This method is denoted as GrW (Gradient times weight) on Figures 3 and 7.

M-FAC Pruner proposed in (Frantar et al., 2021) is a pruner leveraging second order information that doesn't require an explicit construction of Fisher Inverse matrix. Therefore, unlike WoodFisher and oViT that require $O(Bd)$ memory computation and storage cost of this method is constant with respect to the block size and one can take into account correlations between larger groups of weights for free. Following the original paper we chose block size of 2k as the best performing one. However, one can see from Figures 3 and 7 that smaller block sizes turn out to perform better. A possible explanation of this phenomenon is that the Fisher Inverse estimate becomes too noisy and unstable for large blocks.

## I  EXECUTION LATENCY.

In addition to the plot throughput vs accuracy shown in the main part we present in this section execution latency per sample vs latency when running models on the DeepSparse engine. The results are presented on Figure 12.

## J  COMPARISON WITH AC/DC TRAINING

In addition to the sparse training from scratch with periodic updates of the sparsity weights with some saliency criterion for weight elimination and regrowth (Evci et al., 2020) one can consider alternating compressed/decompressed training (AC/DC), proposed in (Peste et al., 2021). Namely one switches between dense stages with standard unconstrained training of the model, and sparse stages when the model is pruned to the target sparsity level and trained with the frozen sparsity mask until the

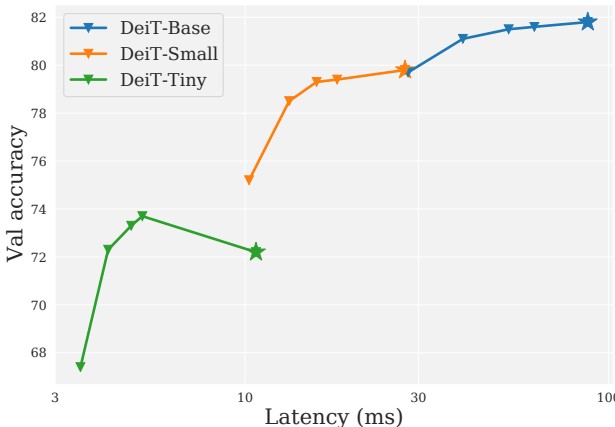

Figure 12: Accuracy vs latency on ImageNet-1k.

beginning of the next dense stage, when the sparsity mask is removed. This procedure produces both accurate dense and sparse models.

Following the original paper we use magnitude pruning as a saliency criterion for weight pruning. The augmentation and regularization pipeline follows the settings from Touvron et al. (2021). All models with AC/DC were trained for 600 epochs in total with first pruning step at epoch 150 followed by 7 sparse stages 25 epochs long each, and 6 dense stages of the same length. The last dense stage lasts 50 epochs and the last sparse is 75 epochs long. Learning rate is gradually decayed from $\eta_{max} = 5 \cdot 10^{-4}$ to $\eta_{min} = 10^{-6}$ with cosine annealing. Initial warm-up phase with linearly increasing learning rate is 20 epochs. We compare AC/DC with oViT models finetuned for additional 100 epochs.

Table 11: AC/DC vs oViT (finetuned for additional 100 epochs) on ImageNet-1k.

| Model | Method | Sparsity (%) | Top1-Accuracy (%) |
|---|---|---|---|
| | oViT | 60 | 79.9 |
| | AC/DC | | **80.4** |
| DeiT-Small | oViT | 75 | **79.0** |
| | AC/DC | | **79.0** |
| | oViT | 90 | **75.8** |
| | AC/DC | | 72.0 |

One can observe that at low sparsity AC/DC achieves higher accuracy for the same sparsity target (even outperforming the dense baseline by 0.6%), whereas for 75% performance of both methods is equal, and oViT outperforms AC/DC at higher sparsity. However, one should note, that oViT uses computational budget (including the training of original model) of 440 epochs for 60% sparsity, 520% for 75% and 700% for 90% vs 600 epochs used in AC/DC.

## K   ONE-SHOT PRUNING OF DETR

The approach presented in the paper is not limited to the image classification task, but can be applied to other computer vision tasks, such as object detection. We chose the DeTR model (Carion et al., 2020) with ResNet50 backbone and ran one-shot pruning procedure with global magnitude, WoodFisher and oViT pruner. Specifically, we pruned all convolutional layers in the CNN backbone expect the first one and all linear projections in transformer encoder and decoder blocks while keeping the detection heads dense. The results are presented in Table 12. Following the standard protocol we used bbox mAP for evaluation. One can observe, that difference between the second order methods and magnitude pruning is very pronounced even for relatively small sparsity of 50%, and oViT outperforms WF pruner.

Table 12: One-shot pruning of DeTR.

| Model | Method | Sparsity (%) | bbox mAP |
|-------|--------|:------------:|:--------:|
|       | Dense  | 0            | 0.42     |
| DeTR  | GM     |              | 0.16     |
|       | WF     | 50           | 0.36     |
|       | oViT   |              | **0.38** |

## L   PROOF OF THEOREM 1

**Theorem 1.** *Let $\mathcal{S}$ be a set of samples, and let $\nabla_{\ell_1}(\mathbf{w}^*), \ldots, \nabla_{\ell_m}(\mathbf{w}^*)$ be a set of gradients with $i \in \mathcal{S}$, with corresponding empirical Fisher matrix $\widehat{\mathbf{F}}^{-1}(\mathbf{w}^*)$. Assume a sparsification target of $k$ weights from $\mathbf{w}^*$. Then, a sparse minimizer for the the constrained squared error problem*

$$min_{\mathbf{w}'} \frac{1}{2m} \sum_{i=1}^{m} \left( \nabla_{\ell_i}(\mathbf{w}^*)^\top \mathbf{w}' - \nabla_{\ell_i}(\mathbf{w}^*)^\top \mathbf{w}^* \right)^2 \text{ s.t. } \mathbf{w}' \text{ has at least } k \text{ zeros}, \quad (10)$$

*is also a solution to the problem of minimizing the Fisher-based group-OBS metric*

$$argmin_{Q,|Q|=k} \frac{1}{2} \cdot {\mathbf{w}_{\mathbf{Q}}^*}^\top \left( \widehat{\mathbf{F}}^{-1}(\mathbf{w}^*)_{[Q,Q]} \right)^{-1} \mathbf{w}_{\mathbf{Q}}^*. \quad (11)$$

*Proof.* We start by examining the unconstrained squared error function in Equation (10), which we denote by $\mathcal{G}$. Clearly, the function $\mathcal{G}$ is a $d$-dimensional quadratic in the variable $\mathbf{w}'$, and has a minimum at $\mathbf{w}^*$. Next, let us examine $\mathcal{G}$'s second-order Taylor approximation around $\mathbf{w}^*$, given by

$$(\mathbf{w}' - \mathbf{w}^*)^\top \left( \frac{1}{m} \sum_{i=1}^{m} \nabla_{\ell_i}(\mathbf{w}^*)^\top \nabla_{\ell_i}(\mathbf{w}^*) \right) (\mathbf{w}' - \mathbf{w}^*), \quad (12)$$

where we used the fact that $\mathbf{w}^*$ is a minimum of the squared error, and thus the function has 0 gradient at it. However, by the definition of the empirical Fisher, this is exactly equal to

$$(\mathbf{w}' - \mathbf{w}^*)^\top \widehat{\mathbf{F}}(\mathbf{w}^*)(\mathbf{w}' - \mathbf{w}^*). \quad (13)$$

The Taylor approximation is exact, as the original function is a quadratic, and so the two functions are equivalent. Hence, we have obtained the fact that, under the empirical Fisher approximation, a $k$-sparse solution minimizing Equation 10 will also be a $k$-sparse solution minimizing Equation 1. However, the question of finding a $k$-sparse solution minimizing Equation 1 is precisely the starting point of the standard OBS derivations (see e.g. (Singh & Alistarh, 2020) or (Kurtic et al., 2022)), which eventually lead to the formula in Equation (11). This concludes the proof. $\square$

## M   PERFORMANCE ON THE "SPARSITY MAY CRY (SMC)" BENCHMARK

Concurrently to our submission, a novel benchmark targeting sparse language models has been proposed. The "Sparsity May Cry" Benchmark (SMC-Bench) (Anonymous, 2023) introduces a set of challenging tasks, spanning from commonsense and arithmetic reasoning to protein termostability prediction and multilingual translation, on which it claims that all existing pruning approaches fail to produce accurate and sparse models, sometimes even at trivial sparsities as low as 5%. As such, it poses an open question on whether the existing datasets and metrics are reliable for evaluation and comparison between different pruning techniques.

We conduct experiments with our proposed oViT pruner on the *hardest benchmark* presented in the SMC-Bench work, according to its authors, on which *all* pruning approaches appear to fail. Namely, the task evaluates commonsense reasoning capabilities of the massive RoBERTa-Large (Liu et al., 2019) model, with 354M parameters, on CommonsenseQA, a question-answering challenge targeting commonsense knowledge (Talmor et al., 2018).

In Figure 13 we present our results on various sparsity targets. More specifically, for 20%, 50%, 60% and 70% targets we simply do one-shot pruning with oViT followed by 5-epochs of fine-tuning. For higher sparsity targets, 80% and 90%, we run gradual pruning with oViT over the span of 20-epochs. As can be seen from the Figure, our proposed oViT pruning approach does not fail even on the hardest

SMC-Bench task and significantly outperforms all other existing techniques. These results also demonstrate that our method, even though developed in the context of vision transformers, produces state-of-the-art results when applied to pruning of large language models in the NLP domain.

Since the exact data of SMC-Bench is not yet released, we sketch our oViT results in Figure 13 and provide exact numbers in Table 13.

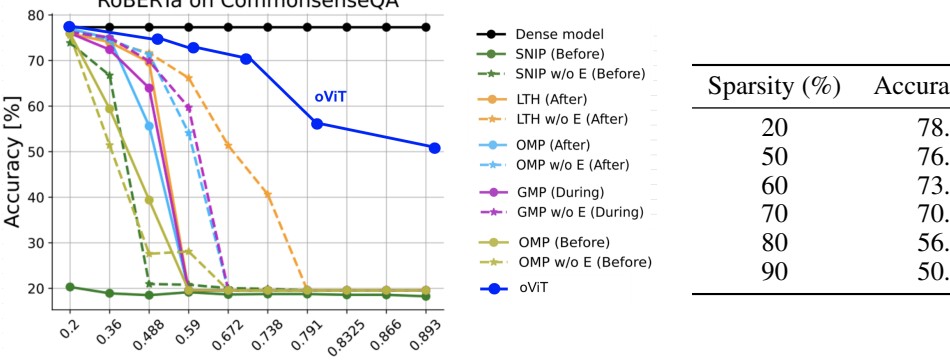

| Sparsity (%) | Accuracy (%) |
|---|---|
| 20 | 78.38 |
| 50 | 76.16 |
| 60 | 73.05 |
| 70 | 70.35 |
| 80 | 56.10 |
| 90 | 50.37 |

Figure 13: Evaluation of various pruning techniques on the challenging SMC-Bench, pruning of the RoBERTa-Large model on the CommonsenseQA task.

Table 13: Sparsity-accuracy results obtained with oViT pruning approach, illustrated in Figure 13.

