# OpenReview forum: "oViT: An Accurate Second-Order Pruning Framework for Vision Transformers"
_ICLR.cc/2023/Conference — Submitted to ICLR 2023_

### Official Review · Reviewer_rP9A · 2022-10-24

**Confidence:** 3
**Correctness:** 4
**Technical Novelty And Significance:** 2
**Empirical Novelty And Significance:** 2
**Recommendation:** 5

**Clarity, Quality, Novelty And Reproducibility:**

The paper is well organized and easy to follow. The novelty should be clarified, compared to the existing OBS based pruning methods. The code is provided which makes the paper easy to reproduce.

**Strength And Weaknesses:**

Strength:
- The paper is well organized and easy to follow.
- The observation on vision transformer pruning is interesting and provides much insight.
- The proposed correlation-aware second-order pruning is reasonable.

Weaknesses or questions:
- Optimal Brain Surgeon [1] is a widely-used second-order technology and has been used in neural network pruning [2,3,4]. This paper is an extension of OBS for vision transformer. The novelty is limited. What's new in the proposed method?
- Could the proposed method be applied on CNN pruning?
- The proposed finetuing recipe is a set of tricks, which is more like engineering contribution.
- The proposed finetuing recipe includes learning rate schedule, regularization, augmentation and an efficient pipeline for sparsity sweeps, The ablation study of these components are not fully conducted.

[1] LeCun Y, Denker J, Solla S. Optimal brain damage[J]. Advances in neural information processing systems, 1989, 2.

[2] Hassibi B, Stork D. Second order derivatives for network pruning: Optimal brain surgeon[J]. Advances in neural information processing systems, 1992, 5.

[3] Hassibi B, Stork D G, Wolff G J. Optimal brain surgeon and general network pruning[C]//IEEE international conference on neural networks. IEEE, 1993: 293-299.

[4] Dong X, Chen S, Pan S. Learning to prune deep neural networks via layer-wise optimal brain surgeon[J]. Advances in Neural Information Processing Systems, 2017, 30.

**Summary Of The Paper:**

The paper focuses on unstructured sparsity of vision transformer. The paper proposes a second-order pruner called oViT, and a set of general sparse fine-tuning recipes. The proposed oViT allows pruning in a single run. The experiments on ImageNet-1K show the effectiveness of the proposed method.

**Summary Of The Review:**

The paper is well written but the novelty should be further clarified, compared to the existing OBS based pruning methods. The experiments on finetuning recipe should be more comprehensive.

---

> ### Author Response · Authors · 2022-11-12
> **Review Response**
>
> Thank you for your feedback! We are glad that you found the paper interesting and well-organized.
> We address your individual questions below:
>
> > What is new in the proposed method?
>
> As formulated in the 1990s, second-order pruning is not applicable to modern neural networks due to extremely high computational and storage cost: storing the Hessian matrix for DeiT-Small would require more than a Petabyte of memory. (OBD/OBS were initially applied to networks with hundreds of weights.)
> Instead, there is a lot of recent work on efficient and accurate approximations, e.g. [L-OBS, NeurIPS17], [WoodFisher, NeurIPS20], and [M-FAC, NeurIPS21]) papers. Coming up with new technical insights in this area seems non-trivial.
>
> Our work is motivated by the observation that all existing second-order pruning methods only consider the diagonal elements of the inverse Hessian when selecting the sets of weights to prune. This ignores the large number of correlations given by the off-diagonal entries, which, intuitively, is very significant for Transformer models. To address this, we provide a new analytical insight (Theorem 1), which demonstrates that these correlations can actually be taken into account accurately and efficiently, by transforming the Fisher weight selection into a sparse regression problem, and solving it efficiently.
>
> Thus, oViT is the first method to efficiently solve for correlations between the weights of a deep network when pruning. At the practical level, this allows us to outperform existing approximations, often by wide margins. Our method does particularly well on ViTs, which exhibit such correlations, and have become very popular recently. Specifically, on ViTs we outperform both existing second-order methods, and the state-of-the-art ViT pruning method [SViTE, NeurIPS21].
>
> > Could the proposed method be applied on CNN pruning?
>
> As stated in our paper, our pruner is not specific to ViTs. It can be applied to any other computer vision architecture. In Appendix D, we present applications of our method to CNN and hybrid Transformer-CNN architectures, where it again outperforms other methods. Moreover, we show that our method is also applicable to detection models (DeTR).
>
> > The proposed finetuing recipe is a set of tricks, which is more like engineering contribution.
>
> It is indeed the case that we complement our method with several practical contributions in terms of efficient finetuning schedules for ViT models. However, we note that this contribution should not be easily dismissed:
>
> * First, please note that finding good training/finetuning recipes for ViTs is extremely complex: as we show, these models require very careful parametrization when fine-tuning, which is not the case for CNNs. This is required for good pruning results, and so our results should be quite useful for future work on compression.
>
> * Second, notice that there is significant work in the community on data-efficient *training* of ViTs, specifically the [DeiT, ICML21] and [“How to train your ViT”, TMLR22] papers. The fact that these papers are highly-cited and appeared in good venues suggests that this work is valuable to the community.
>
> > The proposed finetuing recipe includes learning rate schedule, regularization, augmentation and an efficient pipeline for sparsity sweeps,
> The ablation study of these components are not fully conducted.
>
> We did present partial ablations with respect to the learning rate and augmentation schedules. Specifically, in Subsection 4.2 ( Figure 4 ) we showed that a cyclic-linear schedule produces better results than conventional cosine annealing (see red and green curves). Both these runs use the same augmentation / regularization, so are an ablation on the learning rate schedule. Further, we compared the augmentation / regularization recipe (green curve vs blue curve) with the standard recipe used in DeiT-Small in Figure 4 (bottom). The standard recipe turns out to work better, since the model has more parameters, and sufficient regularization is needed.
>
> If the reviewer is interested in an ablation relative to specific parameters, we would be happy to try to accommodate this.
>
> However, an exhaustive study involving ablations across multiple params - properties of random augmentation, weight decay, hyperparams of the learning rate schedules – would require a very large amount of computational resources, and would completely change the focus of our work. We aimed to study the impact of regularization/learning rate on pruning, and for this we present state-of-the-art results.
>
> We thank you again for your comments, and would be happy to know if we fully addressed your concerns.

---

> > ### Comment · Reviewer_rP9A · 2022-11-22
> > **Thanks for the response**
> >
> > Although some parts of my concerns were addressed by the rebuttal. Some key problems are not solved: 1) The ablation study is not solid, as the proposed finetuing recipe includes a set of tricks, and we cannot figure out what parts are effective. 2) The proposed method can be applied on CNN, so are some new problems in ViT pruning solved?

---

> > > ### Author Response · Authors · 2022-11-22
> > > **Response Part 1: Ablations**
> > >
> > > Thank you for engaging with us, and we are glad we have partly addressed your concerns.
> > > We address your remaining questions in detail below:
> > >
> > > > 1) The ablation study is not solid, as the proposed finetuning recipe includes a set of tricks, and we cannot figure out what parts are effective.
> > >
> > > We believe we provide ablation studies for practically every aspect of our method. Specifically:
> > >
> > >
> > >
> > > * In [Figure 4](https://openreview.net/pdf?id=zYWtq_HUCoi#page=8) we perform ablations on fine-tuning schedules. We perform ablations on the *learning rate sequence,* showing the importance of both the linear learning rate schedule relative to the Cosine LR schedule that is SOTA for ResNets [Kusupati et al., 2020] (see top graph).
> > >
> > > Please note that this graph suggests that Cosine LR methods, which are SOTA for CNNs, are sub-optimal for ViT models.
> > > To further reinforce this point, we provide a similar graph for ViT-Small at [this anonymized link](https://www.dropbox.com/s/68yqrztwptljb1m/gradual_pruning_deit_small_aug_comparison.pdf), showing the same point.
> > > (We had executed this ablation study as well, but did not include it due to lack of space.)
> > > We believe this additional study fully clarifies the impact of the learning rate schedule and augmentation.
> > >
> > > Further, we performed an ablation on *the impact of careful regularization* for good recovery on ViT models (see both top and bottom graphs).  Finally, we perform a full ablation across models and methods, in our fixed finetuning setting in [Table 2](https://openreview.net/pdf?id=zYWtq_HUCoi#page=8).
> > >
> > > Above, GMP uses *exactly the same finetuning as oViT*. One can observe the improvement from our finetuning approach (since sometimes GMP outperforms SViTE!). At the same time, one can also observe the importance of our superior pruning method, as we consistently outperform GMP, often by wide margins.
> > >
> > > * In [Figure 3](https://openreview.net/pdf?id=zYWtq_HUCoi#page=7) we provide an ablation study for *one-shot* pruning, across all known accurate pruners: magnitude, first-order, and several second-order approximations, each with carefully-tuned parameters.
> > > **This figure shows clearly that our method is the best at one-shot pruning, across all known pruning methods, on both ViT and ResNet models, by a wide margin. Please note that this experiment does not include any fine-tuning at all.**
> > >
> > > Also in this figure, please note that using pruners other than oViT, the Visual Transformer drops accuracy at a significantly higher rate than the ResNet, for the same sparsity. Concretely: if the ResNet can be one-shot pruned to 60-70% sparsity using WoodFisher without much loss, the same method starts dropping accuracy significantly even at 40% for the ViT model. This shows that ViTs are much more sensitive to pruning using prior techniques, so they *require* better pruners for good performance. This is what motivated our investigation into more accurate pruners, which led to our oViT pruner method.
> > >
> > > * In [Table 1](https://openreview.net/pdf?id=zYWtq_HUCoi#page=7) we provide an ablation study across different pruners and sparsity levels with a single fine-tuning stage, for 20 epochs, across all methods. This suggests that the superiority of oViT comes from better pruning at every step, which does not disappear after fine-tuning.
> > >
> > > * We emphasize the fact that our appendix contains significantly more ablations:
> > >
> > > 1. In [Appendix G](https://openreview.net/pdf?id=zYWtq_HUCoi#page=17), we show ablations across each individual hyper-parameter of our method: number of gradients (Figure 8), dampening constant (Figure 9) and block size (Figure 10). Each of which is contrasted against WoodFisher.
> > >
> > > 2. In [Appendix D](https://openreview.net/pdf?id=zYWtq_HUCoi#page=14), we show ablations across a wide range of models (EfficientFormer, EfficientNet, etc.) and sparsities, showing the stability of our results.
> > >
> > > 3. Finally, in [Appendix M](https://openreview.net/pdf?id=zYWtq_HUCoi#page=21) we show a comparison against prior pruning methods on the recent and challenging [SMC Benchmark](https://openreview.net/forum?id=J6F3lLg4Kdp). Results suggest that our method clearly outperforms prior methods on this pruning benchmark. In particular, oViT seems to be the first method whose performance does not crash at high sparsities.
> > >
> > > We hope that this clarifies the question of ablations. If the reviewer would like us to perform any specific additional ablation, we would be happy to try to do so in the remaining time.

---

> > > > ### Author Response · Authors · 2022-11-22
> > > > **Response Part 2: Problems in ViT Pruning**
> > > >
> > > >
> > > > > 2) The proposed method can be applied on CNN, so are some new problems in ViT pruning solved?
> > > >
> > > > The main problem in ViT pruning we are solving is the following:
> > > >
> > > > **Prior to our work, the max sparsity reached by the SOTA ViT pruning method, SViTE, published in NeurIPS21, was ~50% sparsity, before significant accuracy loss occurred. This statement is easy to check from the SViTE paper, or from our Table 2.**
> > > >
> > > > This is an important problem for (at least) two reasons:
> > > > * Conceptually, this led to the intuition present in the literature that ViT models must have more parameters than CNNs to achieve good accuracy.
> > > > * Practically, this limitation of existing methods significantly affected the speedups which can be achieved from sparsity for ViTs: computational speedups at 50% sparsity are almost negligible.
> > > >
> > > > Our paper shows that accuracy loss is not inherent for high sparsity on ViTs: we reach 75-80% sparsity or more, across all ViT variants, with less than 1% relative accuracy loss. This shows that ViTs can provide similar accuracy-per-parameter to CNNs, and provide very healthy inference speedups, of 2-3x, as shown in [Figure 5](https://openreview.net/pdf?id=zYWtq_HUCoi#page=8).
> > > >
> > > > We achieve this via:
> > > > 1. An accurate and efficient mechanism which can take weight correlations into account when pruning (oViT, Theorem 1).
> > > > This is a general contribution, and indeed would apply to any model. We believe that this is a strength, not a weakness of our method.
> > > > Specifically, our results (Figure 3, Appendix M) show that this is now the state-of-the-art second-order pruner across all models we have tried (e.g., ViTs, CNNs, DeTR, RoBERTa).
> > > >
> > > > 2. A ViT-specific contribution in terms of highly-efficient and highly-accurate recipes for both sparsity sweeps and  fine-tuning.
> > > > We stress that this contribution is not trivial: for illustration, performing the full sparsity sweep (50, 60, 75, 80, 90% sparsities) using SViTE on the DeiT-Small model requires 5x 600 = 3000 epochs of training on ImageNet, which is approximately 38 days on 4x NVIDIA A6000 GPUs.
> > > > By contrast, our method requires 8.5 days for *the full sweep,* using only *half the resources*, i.e. 2x A6000 GPUs, under exactly the same conditions (and obtains significantly better results).
> > > >
> > > > We therefore believe that both of these findings---a general, SOTA pruner, and specialized recipes that set new SOTA results for ViTs efficiently---should be very useful for the community.

---

### Official Review · Reviewer_rRsE · 2022-10-25

**Confidence:** 5
**Clarity, Quality, Novelty And Reproducibility:** Clarity - good, quality - good, novel…
**Correctness:** 3
**Technical Novelty And Significance:** 2
**Empirical Novelty And Significance:** 3
**Recommendation:** 5

**Strength And Weaknesses:**

Strengths:

- Transformers are popular and pruning them is important. The proposed method is generic and it is not clear how it is specific to transformers.
- Working with saliency metric other than the magnitude is a promising direction. Specially because weight magnitude does not scale for global pruning and requires sensitivity analysis.
- Taylor decomposition based techniques are demonstrated to be better for CNNs, it is good to see this works for transformers as well.
- Proposed method computed saliency and the required weight update.

Weakness/questions:

- What is exact pruning setup? Is it global unstructured sparsity or uniform unstructured sparsity? For the the former one it is clear that GM will not perform well as weights are not scaled with respect to the layer number.
- With all approximation, what is the final equation to compute saliency? authors mention eq 2 in Algorithm 1, but this is the start point not oViT. Sharing code will help to understand the algorithm.
- In the paper authors state and try to answer why pruning transformers is hard. However, pruning of ViT should not be harder as transformers are overparametrized. [1] shows that Diet pruned for structured sparsity of 50% maintains accuracy. For N:M sparsity, for example 2:4, pruning with magnitude gives no accuracy drop.
- Finetuning is an important aspect of the pruning method. Authors tune the recipe, what happens if the standard recipe is used.
- Method is based on the Taylor expansion of the the loss function. The method is inspired by OBS and therefore should be compared directly even if compute time is significant. Having comparison with optimal brain damage will be helpful as well.
- How would the method perform if we use diagonal approximation of the Hessian: $\rho = w_i^2*\Delta L(w_i)^2$ ? This is another popular saliency metric for pruning.
- Transferability of the concluded insights: the experiments are conducted on classification tasks only. While the observations on different tasks maybe different, thus I am not sure whether the extracted insights can be transferred to other tasks or it is only a not general conclusion for classification tasks.
- What is special in the proposed method for transformers? The method seem to be general but title specifically days "for vision transformers"
- Minor: LeCun proposed optimal brain damage (OBD) while Hassibi proposed optimal brain surgeon.


[1] Yang, Huanrui, et al. "Nvit: Vision transformer compression and parameter redistribution." arXiv preprint arXiv:2110.04869 (2021).


**Summary Of The Paper:**

Paper proposes a method to prune neural networks. Most experiment are shown for Transformers, results on CNN are presented in the Appendix. The method is based on second order statistics, Hessian. The method is inspired by Optimal Brain Surgeon (1993) and is the modification of it. Results are presented on Imagenet image classification.



**Summary Of The Review:**

Paper proposes a general framework for pruning neural networks. It is based on iterative saliency pruning. Novelty of the method is in the computation of the saliency metric that is based on Hessian inverse and a modification of OBS (1993). In my understanding, the paper proposes a way to simplify computation of the Hessian inverse, therefore novelty is questionable. Paper lacks comparisons with simpler techniques based on first order Taylor expansion methods that are simpler (gradient times weight), as well as full versions of OBS and OBD. Finally, the method is not specific to Transformers and more comparisons are required to the SOTA in CNN pruning.

---

> ### Author Response · Authors · 2022-11-12
> **Review Response [Part 1]**
>
> Thank you for your interesting and thorough comments! We address each question individually:
>
> > What is exact pruning setup? Is it global unstructured sparsity or uniform unstructured sparsity?
>
> We considered global sparsity (see Section 4), that is, weight importance is computed over the whole model, and sparsity is different in different layers. Unlike in CNNs, the hidden dimension and size of the feature maps is the same in all transformer blocks (for the case of a vanilla transformer), therefore one doesn’t have to account for the different resolutions and widths of layers.
>
> To fully address this, in Appendix C we present a comparison between uniform and global sparsity for the magnitude and the oViT pruner (see Figure 6 in the revision). One can see that the difference between uniform and global sparsity is negligible for magnitude pruning. For the case of oViT, global sparsity performs slightly better. In any case, please note that the difference between the pruners is larger than the difference between the uniform/global sparsity, so the choice of pattern would not affect our conclusions at all.
> In the revised version of the manuscript we added one-shot pruning results of ResNet50, where the difference between uniform and global sparsity is more pronounced. oViT always significantly outperforms the magnitude pruner.
>
> > With all approximation, what is the final equation to compute saliency?
>
> The saliency score for oViT is the same as for the original OBS, or for WoodFisher, i.e. weight squared over inverse diagonal entry. The difference is in the fact that the WoodFisher approach computes saliency score once and *doesn’t account for weight correlations* when removing multiple weights. That is, the change of the Hessian after some of the weights have been eliminated from the model is ignored. On the contrary, oViT proposes an efficient greedy selection and update procedure, such that after every single weight elimination, the remaining weights, the inverse Fisher and the pruning scores are adjusted according to their correlations. This results in significantly higher accuracy.
>
> > Sharing code will help to understand the algorithm.
>
> The full implementation was made available with the original submission, as a docker container. It is accessible as a ZIP file at the top of this page.
>
> > In the paper authors state and try to answer why pruning transformers is hard. However, pruning of ViT should not be harder as transformers are overparametrized. [1] shows that Diet pruned for structured sparsity of 50% maintains accuracy. For N:M sparsity, for example 2:4, pruning with magnitude gives no accuracy drop.
>
> We in fact fully agree with the reviewer’s intuition here, but wish to make some distinctions.
>
> * The vision transformers used in our work have approximately the same number of parameters as the CNN models used in prior work on model compression on ImageNet scale tasks. For instance, DeiT-Tiny is of the same scale as MobileNetV3 Large and DeiT-Small is a bit smaller than ResNet 50 - 22M params vs 25M params. So in that sense they are not “more overparametrized” than prior models.
>
> * However, ViTs are harder to prune, in the sense that they drop more accuracy than ResNets after pruning. The key distinction we wish to make is that the hardness of pruning ViTs is not because of lack of *capacity* (here, we fully agree with the reviewer), but that one needs 1) extremely precise pruners in order not to drop massive accuracy on the pruning step; 2) an effective and cost-efficient finetuning strategy in order to recover lost accuracy. We provide solutions for both problems in our work.
>
> * Our claim of hardness stems from the fact that the prior state-of-the-art for ViT pruning (SViTE, published in NeurIPS21), which uses a state-of-the-art CNN pruning method with significant training cost, can only prune to ~50% sparsity before significantly dropping accuracy, In addition, other baselines (GMP, WF) do not do well either when run directly in their original setting. (Notice that we run them with our own specific augmentation/learning rate schemes, for fairness.)
>
> * Concerning the NViT work, notice that they do not only prune the model but also change the architecture:  dimensions of keys, values, number of heads, such that the model obtained has higher performance with the same amount of compute. This work proposes a kind of neural architecture search, which is a practical approach for model optimization but orthogonal to our line of work on unstructured sparsity. In particular, NAS tends to be very computationally-intensive, whereas we provide much more efficient compression strategies, which lead to non-trivial speedups (Figure 5). We will further clarify this in the paper.
>
> * Regarding 2:4 pruning, please notice that our fine-tuning recipe is much (5-10x) shorter than the full-retraining approach usually employed to recover accuracy after 2:4 pruning.
>
> <Our response continues onto Part 2.>

---

> ### Author Response · Authors · 2022-11-12
> **Review Response [Part 2]**
>
> > What happens if the standard training recipe is used?
>
> We examined this point in Figure 4 in the original submission, where we showed that using the SOTA training recipe is suboptimal for recovering accuracy on DeiT-Tiny, motivating our use of lower regularization/augmentation. Moreover, in the same figure, we have shown that a SOTA learning rate schedule for CNNs is grossly suboptimal  for ViTs, relative to the cyclic learning rate schedule we propose in our work.
>
> > Comparisons with OBS and OBD:
>
> Comparing directly with OBS is impossible: the method proposes to store the entire Hessian for the network, which is Petabytes in size even for the smaller ViT models. Even the exact computation of OBD (Hessian diagonal) is extremely expensive, since it requires to re-run backpropagation for every weight in the model.
>
> However, we have compared against the best-known approximate implementations of OBD/OBS, which are WoodFisher (blocked) and M-FAC (global Hessian approximation). In addition, we have implemented and compared against gradient * weight (marked as GrW), and OBD/Fisher diagonal (marked as WF-1). The results are presented in Appendix C of the revision. We note that oViT outperforms all prior approximations.
>
> > I am not sure whether the extracted insights can be transferred to other tasks or it is only a not general conclusion for classification tasks.
>
> To address this, we have added experiments with DeTR (object detection model) in the revision(see Appendix J), and on RoBERTa-Large (language model, SMC Benchmark) in Appendix M.
>
> > What is special in the proposed method for transformers?
>
> The *pruning* algorithm we propose is not specific for Vision Transformers and can be applied to any computer vision architecture and even different domains (e.g. NLP) without significant modifications. However, our method was specifically motivated by the hardness of obtaining accurate sparse ViTs, and our contributions regarding data-efficient finetuning and efficient sparsity sweeps are specifically aimed at ViTs.
>
> > In my understanding, the paper proposes a way to simplify computation of the Hessian inverse, therefore novelty is questionable.
>
> The problem of estimating the Hessian inverse for deep nets has a very long history, and several approximations have been proposed, for instance OBD/OBS, the entire K-FAC line of work, or the more recent [L-OBS, NeurIPS17], [WoodFisher, NeurIPS20], and [M-FAC, NeurIPS21]) papers. One could therefore argue that it is non-trivial to come up with new technical insights in this area.
>
> Specifically, in terms of technical novelty, our approach is the first to efficiently take into account correlations between weights when selecting a pruning mask: for computational reasons, all existing second-order methods only consider the diagonal elements of the inverse Hessian when selecting the sets of weights to prune, which ignores the large number of correlations given by the off-diagonal entries. Our approach is based on a new analytical insight (Theorem 1), which demonstrates that these correlations can actually be taken into account accurately and efficiently, by transforming the Fisher weight selection into a sparse regression problem, which we solve efficiently.
>
> At the practical level, this allows us to outperform existing approximations, often by wide margins. This was shown in our original submission, where we compared against the best-performing second-order method on ViTs (WoodFisher). We reinforce this in the revision (Appendices C and D), where we show that we outperform all simpler approximations (diagonal, gradient * weight, M-FAC, and WoodFisher).
>
> > Paper lacks comparisons with simpler techniques based on first order Taylor expansion methods that are simpler (gradient times weight), as well as full versions of OBS and OBD.
>
> Our original submission compared against the best prior pruning technique (WoodFisher), which is known to significantly outperform simpler approaches (see Singh et al, NeurIPS20).
> Yet, we acknowledge this comment. To address it, we have implemented and compared against *all existing approaches*: OBD (diagonal approximation), gradient * weight, OBS (WoodFisher), and global Hessian/M-FAC.
>
> The full results are provided and discussed in Appendix C of our revision.
>
> > Finally, the method is not specific to Transformers and more comparisons are required to the SOTA in CNN pruning.
>
> In Figure 3 of our revision, we have added a full comparison between SOTA CNN pruning methods–WoodFisher and M-FAC—on the compression of a ResNet50 model. The Appendix contains a full discussion. Again, results show that our approach outperforms all other previous pruners, even in this setup.
>
> We thank you again for your feedback, which helped improve our paper. We believe that the above results substantively address your comments, and would be happy to continue the discussion further.

---

> ### Author Response · Authors · 2022-11-20
> **Evaluation on transfer learning**
>
> To further address the concern on the generality of the results obtained we have evaluated the original and sparsified with oViT models on set of downstream tasks following the linear finetuning setting from the [paper](https://arxiv.org/abs/2111.13445) [1]. I.e the backbone was frozen and only classification head is trained on top of dense or pruned model features.
>
> One can observe that sparse models get almost the same accuracy as the original model even at high sparsities which means that the features are still good enough for high performance on the downstream tasks.
>
> |    Model   | Sparsity | Speed-Up | CIFAR100 | FGVC Aircraft | Oxford Pets |  DTD | Caltech 101 | Caltech 256 | Stanford Cars | Food 101 |
> |:----------:|:--------:|:--------:|:--------:|:-------------:|:-----------:|:----:|:-----------:|:-----------:|:-------------:|:--------:|
> | DeiT-Small |     0    |     1    |   75.1   |      37.4     |     93.4    | 69.8 |     91.1    |     83.8    |      51.2     |   72.1   |
> |            |    50    |   1.54   |   74.5   |      34.8     |     93.5    |  70  |     90.4    |     83.6    |      50.8     |   71.8  |
> |            |    75    |   2.08   |   74.1   |      37.1     |     93.7    |  69  |     90.2    |     83.4    |      48.4     |   71.1   |
> |            |    90    |   2.69   |   71.3   |      34.9     |     92.8    | 67.3 |     89.9    |     81.5    |      46.3     |   68.4   |
>
> Speed-up in the table above is relative increase in throughput compared to the dense model.
>
> [1] Iofinova, Eugenia, et al. "How Well Do Sparse ImageNet Models Transfer?." Proceedings of the IEEE/CVF Conference on Computer Vision and Pattern Recognition. 2022.

---

> ### Author Response · Authors · 2022-12-13
> **Feedback on Our Response**
>
> Dear Reviewer,
>
> We believe that we addressed your questions in a substantive manner, in particular by providing additional experimental data on different tasks (object detection, language modelling, transfer learning) and models (DETR, RoBERTa), as well as clarifying the pruning setup and the relationship to overparametrization.
>
> We would therefore really appreciate it if you could kindly provide feedback on our response, or let us know if there are further questions.
>
> Best regards,
>
> The authors

---

### Official Review · Reviewer_TCn5 · 2022-10-25

**Confidence:** 3
**Clarity, Quality, Novelty And Reproducibility:** See Strength And Weaknesses
**Correctness:** 3
**Technical Novelty And Significance:** 2
**Empirical Novelty And Significance:** 2
**Recommendation:** 6

**Strength And Weaknesses:**


Strength:

-The authors conduct extensive experiments to validate the effectiveness of this method. The experiments are conducted with multiple backbones including ViT, XCiT, Swin-Transformer, e.t.c.

-This paper is well-written and easy to follow.

Concerns:

-Using second-order information to prune neural network is a common strategy. It has been widely used in spare convolutional neural networks. The work brings these methods to vision transformer. Could the second-order method for CNN be used in vision transformer? The authors are required to clarify the contribution and novelty about the method transferring.

- It is interesting that the unstructured pruning can achieve practical acceleration. However, only the speed of proposed method is shown in Table 5. How about the competeting method? Compared with other methods, could the proposed still show superiority with similar practical speeds?  Besides, it is better to illustrate how to achieve practical acceleration on the a CPU platform more detailedly.



**Summary Of The Paper:**

This paper proposes a new weight sparsification for vision Transformer. It considers second-order information to recognize redundant weights. By investigating various strategies, the proposed method can acheive high sparsity levels with low impact on accuracy.

**Summary Of The Review:**

See Strength And Weaknesses

---

> ### Author Response · Authors · 2022-11-12
> **Review Response**
>
> Thank you for your comments! We address each one of your questions individually:
>
> >  Could second-order methods for CNNs be used in vision transformers?
>
> Yes, second-order methods for CNNs can be used for ViTs. As stated at the beginning of Section 4, we have implemented all leading second-order CNN pruners, tested them on ViTs, and selected the one that worked best (WoodFisher, marked as WF) for the comparisons in our paper.
>
> To fully clarify this concern, we have now added a full comparison for one-shot pruning, relative to all recent second-order methods. This can be found in Figure 6, Appendix C, or for simplicity at this [anonymized link](https://www.dropbox.com/s/zfycvrldviv5fq0/one_shot_deit_tiny_method_comparsion.pdf).
>
> The results clearly show that oViT performs better than all prior methods in terms of model quality after pruning, for both layer-wise uniform (U) and global (G) pruning. Please see Appendix C in the revision for a full description of the experimental setting.
>
> > The authors are required to clarify the contribution and novelty about the method transferring.
>
> Relative to existing second-order methods (M-FAC, WoodFisher, etc.), our contribution is a new efficient approximation of second-order pruning statistics, based on a new theoretical insight (Theorem 1) which allows us to re-formulate the multi-weight pruning problem as a variant of sparse regression. oViT outperforms all prior second-order methods precisely because it accounts for weight correlations during the pruning mask selection. Importantly, Theorem 1 enables us to do this very efficiently.
>
> This correlation-aware pruning is especially-useful in the case of vision transformers: these models can really benefit from better pruning, since they are very “sensitive” to single compression steps. At the same time, in our paper we have shown that the method also works very well for CNNs (see Appendix D for results on ResNet and EfficientNetV2, where we again outperform prior methods), and for other tasks (see Appendix J for DeTR results, and M for RoBERTa-Large results).
>
> > Only the speed of the proposed method is shown in Table 5. How about the competing method?
>
> The practical speedup at a given level of sparsity is very similar across different pruning methods, since the per-layer sparsities are not dramatically different. Here, the benefit of oViT is that it is significantly more accurate for the same sparsity level, which leads to an improved accuracy-latency trade-off. In other words, sparse models produced by other pruning methods will reach similar execution latencies but have lower model accuracy.
>
> We illustrate this in [this anonymized figure](https://www.dropbox.com/s/mbihm9ofi8d3q8d/model_throughputs_different_methods.pdf), which shows accuracy-vs-real-speedups for oViT, WoodFisher (WF) and Global Magnitude (GM), for DeiT-Tiny and Small, in a similar setting to that used in our submission.
>
> > Besides, it is better to illustrate how to achieve practical acceleration on a CPU platform more detailedly.
>
> The speedup is achieved via sparsity-aware Conv/GEMM kernels supported by the DeepSparse engine, which is freely-available for researchers and academics, and supports both sparse ViTs and CNNs. More precisely, the Pytorch model can be exported to ONNX, after which it can be directly run with speedup in DeepSparse. For more information on CPU acceleration of unstructured sparse models see for example [ Elsen et al., https://arxiv.org/abs/1911.09723 ] or [Kurtz et al., https://proceedings.mlr.press/v119/kurtz20a.html ].
>
> Thank you again for your comments. We would be happy to know if we have fully addressed your concerns.

---

### Author Response · Authors · 2022-11-12
**General Response**

We thank all reviewers for their very useful feedback, and take the chance to address their questions and comments.

Together with our response, we have also submitted a revision of our paper, which addresses all reviewer comments. Thus, our response will be split into two parts:

* A general part (here) which addresses common points in the reviews, and discusses the main changes in the revision.
* Detailed individual responses to each of the reviews in turn.

---

We now address the main common questions:

1. The paper’s contribution, in particular relative to prior work on second-order methods

Our work provides a new and general second-order method for pruning, which provides state-of-the-art results when pruning the very popular but hard-to-compress ViT models, more than doubling sparsity for the same accuracy level relative to the prior SOTA [SViTE, NeurIPS21]. We complement this method with efficient finetuning schedules, experiments on CNN and hybrid architectures, and show that the method can lead to practical speedups.

At the technical level, we are motivated by the observation that all previous second-order pruning methods do not consider correlations between weights when selecting which weights to remove. This ignores the weight correlations given by the off-diagonal entries in the Hessian, which are intuitively very significant for Transformers. Our work addresses this issue via a new theoretical insight, formalized in Theorem 1: we show that these correlations can actually be taken into account accurately and efficiently, by transforming weight selection into a sparse regression problem. We then solve this sparse regression problem efficiently.

The resulting pruning algorithm is new, and, as we showed in the original submission and reinforce in the revision, outperforms all prior approaches, at similar computational cost. This is true both on ViTs, which are our main focus, but, also on ResNets/EfficientNets, and, as we show in the revision, also for the DeTR detection model.

Specifically, to fully address reviewer comments, we have extended our experimental validation to compare against all reasonable existing approaches: magnitude, gradient * weight, OBD (diagonal approximation), OBS (WoodFisher), and global Hessian/M-FAC, on both ViTs and ResNets. Results show that oViT outperforms all these approaches for both uniform per-layer and global pruning.

2. Is the method specific to ViTs?

Our pruning approach can be directly applied to any model. In the revision, we show that it outperforms all alternative pruners on ResNet50, a standard CNN.

However, our method is particularly effective for vision transformers, given that these models are extremely sensitive to weight pruning. Moreover, we complement our algorithmic contribution with a ViT-specific pruning framework, which provides computationally-efficient pruning for these models, while setting SOTA sparsity results.

---

Specifically the revision contains the following major updates:

* A complete comparison against all other pruning approaches (magnitude, gradient * weight, OBD (diagonal approximation), OBS (WoodFisher), and global Hessian/M-FAC) for both uniform and global pruning, on both ViT and ResNet models (Appendices C and D).

* Results on pruning a detection model (DeTR), showing that our method works in this setting as well, outperforming all alternatives (Appendix J).

* Several clarity improvements meant to address reviewer comments.


We thank the reviewers again for their work, and look forward to their feedback.

With best regards,

The oViT authors

---

### Author Response · Authors · 2022-11-19
**Final Revision Summary**

Dear Reviewers and ACs,

We outline some interesting additions present in our latest revision:

* As suggested by reviewers, we present a detailed comparison between magnitude, first-order, and second-order pruning methods in the one-shot setting for both ViTs (DeiT-Tiny) and CNNs (ResNet50-D). Results, shown in [Figure 3](https://openreview.net/pdf?id=zYWtq_HUCoi#page=7), clearly show that oViT outperforms all existing methods, by a wide margin.

* To further reinforce this point, we show that oViT also provides state-of-the-art results on the recently-proposed SMC sparsity benchmark, on its most challenging task. The results in [Figure 13](https://openreview.net/pdf?id=zYWtq_HUCoi#page=22) show that oViT  outperforms existing methods, by a wide margin, in this challenging gradual pruning scenario, on a massive language model (RoBERTa-Large, 345M parameters).

* To address remaining points, we have also made careful clarifications with respect to the writing of the paper, especially emphasizing the contribution relative to existing second-order methods, and specificity to ViTs.

Together with our [original revision](https://openreview.net/forum?id=zYWtq_HUCoi&noteId=0CQirNbrXy) and review responses, we believe we have addressed all the reviewers' comments in a substantive manner.

We sincerely hope that the reviewers will examine these improvements, and that they will engage with us during the second part of the discussion period.

Best regards,

The Authors

---

### Decision · Program_Chairs · 2023-01-20

**Decision:**

Reject

**Justification For Why Not Higher Score:**

Two out of three reviewers were concerned about the novelty, comparison experiments, and finetuning tricks.

**Justification For Why Not Lower Score:**

Lowest already

**Metareview: Summary, Strengths And Weaknesses:**

Three experts reviewed the paper. The last two reviewers were concerned about the paper's novelty and comparison with competing methods. The first reviewer raised questions about baselines and was satisfied with the authors' rebuttal. However, the rebuttal did not address the reviewers' concerns about novelty, comparison experiments, and the finetuning "tricks". Hence, the decision is **not** to recommend acceptance.